# Ribosome surface properties may impose limits on the nature of the cytoplasmic proteome

Paul E Schavemaker[1†], Wojciech M Śmigiel[1†], Bert Poolman[2,3]*

[1]Department of Biochemistry, University of Groningen, Groningen, Netherlands; [2]Groningen Biomolecular Sciences and Biotechnology Institute, University of Groningen, Groningen, Netherlands; [3]Zernike Institute for Advanced Materials, University of Groningen, Groningen, Netherlands

**Abstract** Much of the molecular motion in the cytoplasm is diffusive, which possibly limits the tempo of processes. We studied the dependence of protein mobility on protein surface properties and ionic strength. We used surface-modified fluorescent proteins (FPs) and determined their translational diffusion coefficients (*D*) in the cytoplasm of *Escherichia coli*, *Lactococcus lactis* and *Haloferax volcanii*. We find that in *E. coli* D depends on the net charge and its distribution over the protein, with positive proteins diffusing up to 100-fold slower than negative ones. This effect is weaker in *L. lactis* and *Hfx. volcanii* due to electrostatic screening. The decrease in mobility is probably caused by interaction of positive FPs with ribosomes as shown in *in vivo* diffusion measurements and confirmed *in vitro* with purified ribosomes. Ribosome surface properties may thus limit the composition of the cytoplasmic proteome. This finding lays bare a paradox in the functioning of prokaryotic (endo)symbionts.

DOI: https://doi.org/10.7554/eLife.30084.001

*For correspondence:
b.poolman@rug.nl

†These authors contributed equally to this work

Competing interests: The authors declare that no competing interests exist.

## Introduction

Many processes in biological cells depend on interactions between macromolecules (proteins and nucleic acids) and thus on the ability of these macromolecules to find each other by translational diffusion. This is especially important in prokaryotes because of the virtual absence of active mechanisms of cytoplasmic transport. It is clear that macromolecules need to diffuse for cells to function. To what extent the actual rate of this diffusion matters depends on the process under consideration and is in many cases unknown. For Brownian diffusion the rate of movement is characterized entirely by the diffusion coefficient, *D*. The exact value of the diffusion coefficient is important to the rate of a process only if it is diffusion limited, e.g. if the necessary conformational changes in an enzyme are faster than the diffusion of reactants. Arbitrarily lowering a diffusion coefficient, e.g. by osmotic stress, can make a process diffusion limited. Examples of diffusion-limited processes are binding of tRNA complexes to the ribosome, which leads to limitation in cell growth (*Klumpp et al., 2013*); and the binding of barstar to barnase, which we know to be diffusion limited because the proteins are designed to have an increased association rate by electrostatic interactions (*Vijayakumar et al., 1998*). Because protein diffusion is influenced by the environment, we need to determine diffusion coefficients in the context of the cell.

The cytoplasm of cells is not only crowded with macromolecules (*van den Berg et al., 2017*) but also consists of various types of nucleic acids and >1000 types of protein (see proteome analysis below); though only 50 protein types make up 85% of the cytoplasmic proteome of *E. coli* (*McGuffee et al., 2010*). Various studies report on the presence of weak and nonspecific interactions between these components. NMR studies on proteins, either in the *E. coli* cytoplasm

**eLife digest** Cells contain many proteins that constantly move about within the cell to carry out tasks that keep the cell running. A protein of average size makes contact with all the other proteins in a typical bacterial cell in a few seconds. This moving about mixes the cell's contents, which is crucial for its survival and reproduction.

Proteins can group together, or with other molecules, to form bigger units. This grouping together depends on the properties of protein surfaces; for example, opposite electrical charges on the surfaces of proteins can cause them to group together. Grouping together proteins causes them to move around cells more slowly. Indeed the ribosome, a protein unit that constructs new proteins and is found in every known species, moves over a hundred times more slowly than the average protein.

Finding out how the total electric charge inside the cells of different species affects the mobility of proteins would give us a better idea of how fast tasks can be carried out in cells. Schavemaker, Śmigiel et al. now show that protein surface charge matters a great deal in the gut microbe *Escherichia coli*. In these cells, negatively charged and neutral proteins move about rapidly whereas positively charged proteins move up to 100 times more slowly.

In the salt-loving microbe *Haloferax volcanii* the positive proteins move relatively fast, but still more slowly than negatively charged and neutral proteins. In all likelihood this is because the protein charges are shielded from each other by large amounts of small charged molecules (which come from salts) in the *Hfx. volcanii* cells.

Schavemaker, Śmigiel et al. suggest that positively charged proteins slow down because they bind to negatively charged ribosomes. Because ribosomes are found in all living cells, understanding how they affect how other proteins move around the cell has a wide range of possible applications. For example, biologists and biotechnologists often produce proteins in *E. coli* for convenient study. Yet very positively charged proteins may bind to ribosomes in *E. coli*, causing experiments to fail. Using cells that shield charges better, such as *Hfx. volcanii* or *Lactococcus lactis*, could solve this issue.

DOI: https://doi.org/10.7554/eLife.30084.002

(*Crowley et al., 2011*; *Ye et al., 2013*) or cell lysates (*Latham and Kay, 2013*), reveal that there are weak interactions between *E. coli* proteins and proteins cytochrome *c*, ubiquitin, and calmodulin. In a computational study on protein interactions it was found that in *E. coli* more highly expressed proteins are constrained in evolution to be less sticky (*Levy et al., 2012*), suggesting that nonspecific interactions are common and consequential. The transient macromolecular interactions *in vivo*, resulting from molecular evolution, are referred to as the quinary structure of proteins (*McConkey, 1982*), and we discriminate these from the more generic nonspecific interactions that occur between molecules without coevolved interfaces.

In this study we set out to study the diffusion coefficients of proteins as a function of their surface properties and thus probe the boundary conditions for the generic nonspecific interactions. Our interest in this was piqued by four datasets from the literature. The first is the scattering of diffusion coefficients in the *E. coli* cytoplasm around a common downward trend when they are plotted against protein molecular weight; the dataset suggests that not only size (and shape) matter (*Mika and Poolman, 2011*). Second, the diffusion coefficient of GFP is faster in the cytoplasm of osmotically-adapted *E. coli* cells than in osmotically-upshifted cells, even at similar cytoplasmic macromolecule volume fraction (*Konopka et al., 2009*). Third, the diffusion coefficient of GFP drops much faster with cell volume (after an osmotic upshift) in *Lactococcus lactis* than in *E. coli* (*Mika et al., 2014*). Fourth, the slowing of diffusion in metabolic energy-depleted cells suggests changes in the dynamic structure of the cytoplasm (*Parry et al., 2014*; *Joyner et al., 2016*; *Munder et al., 2016*). In all four cases differential interactions of proteins with their surroundings may play a role, which are grounded in the surface properties of the macromolecules. Besides (possibly) giving insight into these four phenomena, studying the dependence of mobility on protein surface properties adds to our general quantitative understanding of diffusion; complementing studies on the relation between diffusion coefficients and protein size (*Mika and Poolman, 2011*;

*Mika et al., 2014*; *Kumar et al., 2010*; *Nenninger et al., 2010*; *Mika et al., 2010*), diffusion coefficients and macromolecular crowding (*Konopka et al., 2009*; *Mika et al., 2014*; *Mika et al., 2010*; *Konopka et al., 2006*; *van den Bogaart et al., 2007*), and the dynamic structure of the cytoplasm (*Spitzer and Poolman, 2009*; *Spitzer and Poolman, 2013*).

Here, we use a set of GFP variants with a net charge that ranges from −30 to +25; we also studied two variants of +11 GFP that differ in the distribution of the charge over the surface. All diffusion coefficients were determined by fluorescence recovery after photo-bleaching (FRAP). We study these proteins in the bacteria *Escherichia coli* and *Lactococcus lactis* and the archaeon *Haloferax volcanii*. These three organisms differ in their cytoplasmic ionic strength as shown by measurements on the dominant cation, $K^+$: *E. coli* (0.2 M) (*Shabala et al., 2009*), *L. lactis* (0.8 M) (*Poolman et al., 1987*) (note: *L. lactis* used to be called *Streptococcus cremoris*), and *Hfx. volcanii* (2.1 M) (*Pérez-Fillol and Rodríguez-Valera, 1986*); these values are dependent on environmental conditions, but the differences in potassium ion concentration likely report the differences in ionic strength in these prokaryotes. The difference in ionic strength between *E. coli* and *L. lactis* is also reflected in the higher turgor pressure of *L. lactis* (*Mika et al., 2014*).

## Results

### GFP net charge affects its diffusion coefficient in *E. coli*

We performed fluorescence recovery after photo-bleaching (FRAP; see *Figure 1a,b*) to determine the diffusion coefficients of surface-modified variants of GFP in the *E. coli* cytoplasm. We determined the diffusion coefficient of the −30, −7, 0, +7, +11b, +15 and +25 variants of GFP; see *Figure 1c* for structural models. The numbers indicate the net charge; the 'b' in +11b GFP refers to the distribution of the charge over the surface and will be discussed in more detail below.

For each variant we measured the diffusion coefficients on cells from at least three independent cultures, and for each cell we obtained a single diffusion coefficient. For each GFP variant we plotted the histogram of diffusion coefficients over the population of cells (*Figure 1d*). The −30, −7 and 0 variants of GFP all have the same mean diffusion coefficient of 10–11 $\mu m^2$/s (for table of mean diffusion coefficients see *Supplementary file 1B*). At +7 GFP the diffusion coefficients start to drop, down to a mean value of 0.14 $\mu m^2$/s for +25 GFP.

For the +15 and +25 GFP variants we observed heterogeneous fluorescence in some cells. This ranged from a somewhat higher fluorescence at the poles to a clear ring around the outskirts of the cells (*Figure 2—figure supplement 1a*). For +15 GFP we were able to get rid of the heterogeneities by inducing for a shorter period of time. For +25 GFP, we excluded cells with extreme heterogeneities (see *Figure 2—figure supplement 1a*). The cells with only slightly inhomogeneous fluorescence had similar diffusion coefficients to cells with homogeneous fluorescence and were included in the data (*Figure 2—figure supplement 1b*). The heterogeneities are probably due to exclusion of large complexes (+25 GFP forming clusters of ribosomes) from the nucleoid (see below).

Why does the diffusion coefficient drop with net positive charge? The first thing to realize is that almost nothing in the cytoplasm of the *E. coli* cell truly stands still. The membrane rearranges itself continuously and the DNA has some diffusive motion and rearranges itself during the cell cycle. If +25 GFP would stick to another average sized protein it would not move more slowly than the combination of the two can diffuse. If +25 GFP were to bind some bigger complex, like β-galactosidase (~500 kDa) with a diffusion coefficient of ~1 $\mu m^2$/s, it would diffuse with a similar rate as β-galactosidase.

Based on a census of elements present in *E. coli* cells this means that for +25 GFP to diffuse with a diffusion coefficient as low as 0.14 $\mu m^2$/s it needs to bind to membrane proteins, DNA and/or ribosome-mRNA complexes. For membrane proteins with 12–14 transmembrane helices in *E. coli* and *L. lactis*, *D* is 0.02–0.03 $\mu m^2$/s (*Mika et al., 2014*; *Kumar et al., 2010*). Describing the motion of DNA with a diffusion coefficient is somewhat of a stretch, as its parts do not move freely, but apparent values of 0.000035–0.00007 $\mu m^2$/s have been reported (*Reyes-Lamothe et al., 2008*). In fast growing cells, we expect ribosomes and mRNA to be associated most of the time (*Sanamrad et al., 2014*), and *D* is 0.04 $\mu m^2$/s when a one-component model is used for fitting the data (*Bakshi et al., 2012*). In another study free and bound ribosomes were discriminated and *D* values of 0.40 (~15%) and 0.055 $\mu m^2$/s (~85%) were found (*Sanamrad et al., 2014*). So membrane proteins, DNA and

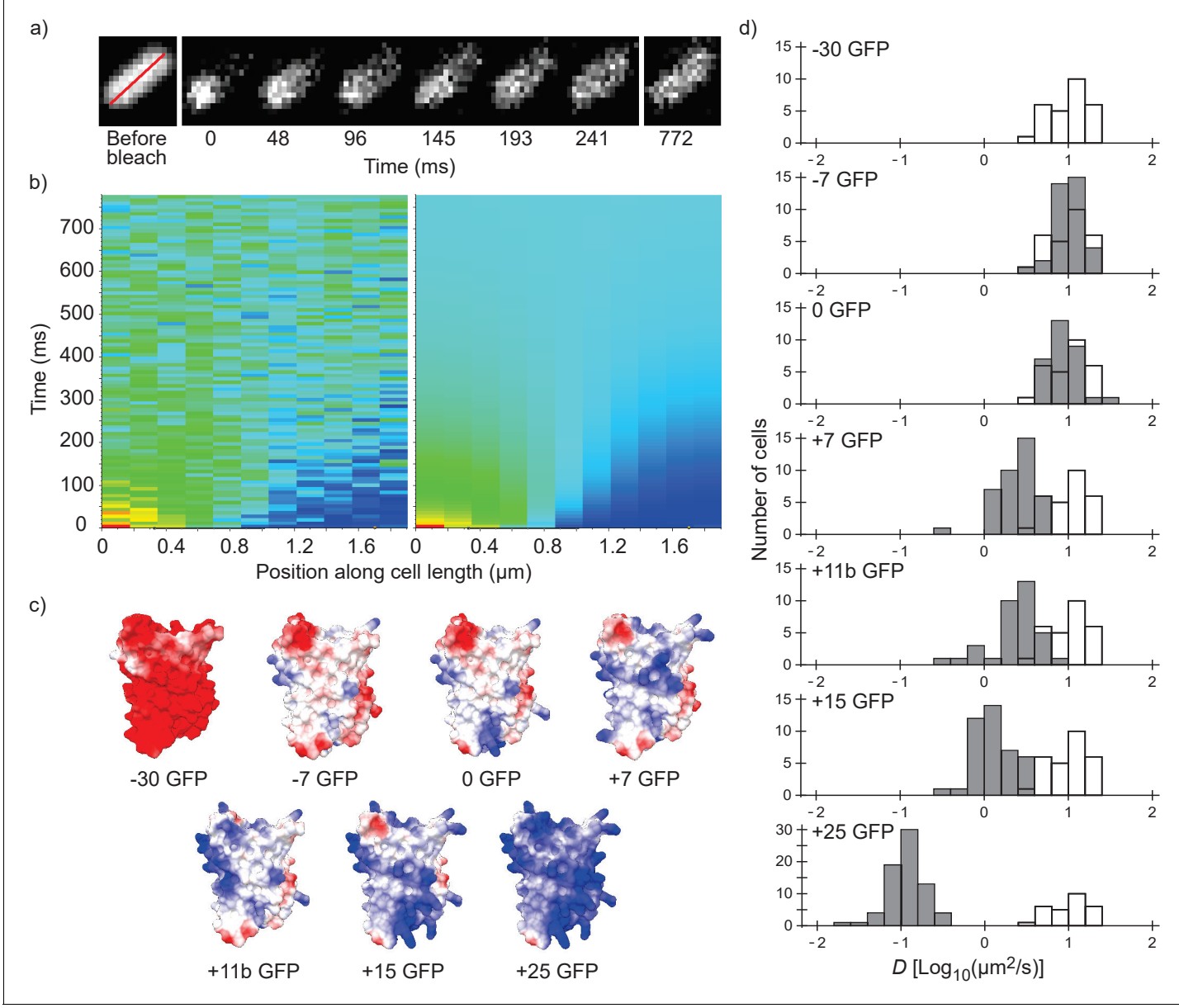

**Figure 1.** Illustration of the fluorescence recovery after photobleaching (FRAP) method, models of GFP variants and histograms of diffusion coefficients of surface modified variants of GFP in *Escherichia coli*. (a) Data from a FRAP experiment. The zero time point is recorded immediately after the bleach. The red line marks the region along which the recovery is analyzed. (b) Fluorescence intensity along the red line in time, for data (left) and the fit to that data (right). The data is fitted with a numerical variant of the 1D diffusion equation. From the fit we obtain the diffusion coefficient. (c) Structural models of the surface-modified GFP variants, based on the superfolder GFP structure (PDBID:2B3P). The colors indicate the charge. The images were generated using UCSF Chimera (*Pettersen et al., 2004*) and MSMS package (*Sanner et al., 1996*). (d) Histograms of diffusion coefficients of GFP variants in *E. coli* over a population of cells. For comparison, the histogram for the −30 GFP variant is shown in white in every plot. *P*-values are reported in **Supplementary file 1C**.

DOI: https://doi.org/10.7554/eLife.30084.003

ribosomes-mRNA all have diffusion coefficients low enough to cause the drop in mobility of +25 GFP. Fluorescence images show that most, if not all, fluorescence is located in the cytoplasm, leaving DNA and/or ribosome-mRNA as the most likely (major) binding partners. We cannot rule out that some of the GFP binds to the membrane but we did not find evidence for it in our images of *E. coli* cells.

# Cytoplasmic ion concentrations counteract the drop in diffusion coefficient

We also determined diffusion coefficients of the GFP variants in *L. lactis* and *Hfx. volcanii* (*Figure 2* and *Supplementary file 1B*). For *L. lactis* the mean diffusion coefficient of −7 GFP (6.2 µm²/s) is lower than for *E. coli* (10 µm²/s). However, *D* drops less with positive net charge so that for +25 GFP the mean diffusion coefficient is higher in *L. lactis* (0.61 µm²/s) than in *E. coli* (0.14 µm²/s). This can be explained by the higher cytoplasmic ionic strength of *L. lactis*, reducing the affinity of positive GFPs to hypothetical negatively-charged binding partners. For *Hfx. volcanii* the drop in diffusion coefficient is even less steep, with the mean diffusion coefficient dropping from 5.5 µm²/s, for −7 GFP, to 1.9 µm²/s, for +25 GFP. The shallower drop in diffusion coefficient with net positive charge as compared to both *L. lactis* and *E. coli* can again be explained by a difference in ionic strength. Another possible contribution to the high +25 GFP diffusion coefficient in *Hfx. volcanii* is the

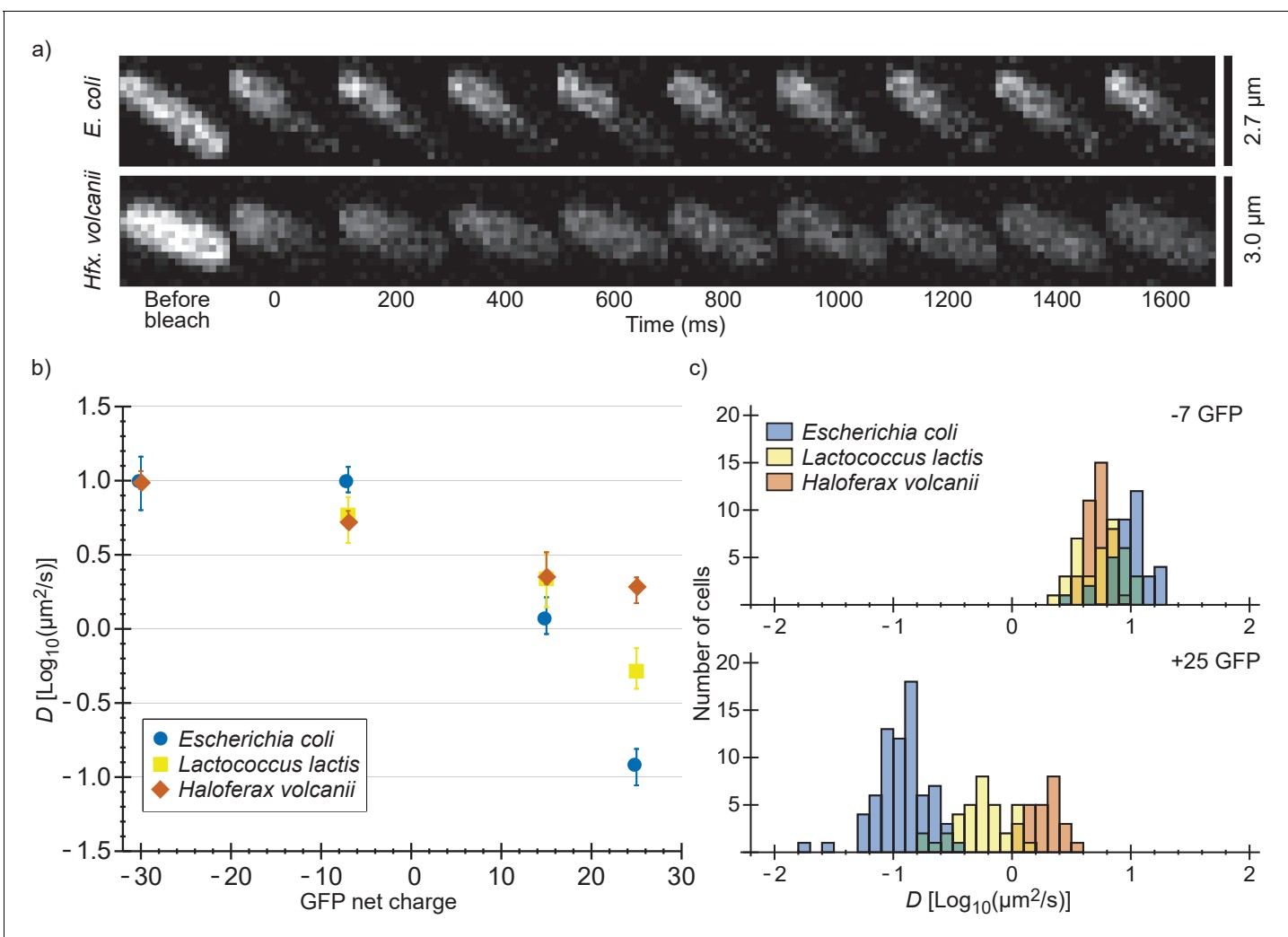

**Figure 2.** Comparison of diffusion coefficients of surface-modified variants of GFP in *E. coli*, *L. lactis* and *Hfx. volcanii*. (a) Example FRAP data for *E. coli* and *Hfx. volcanii* cells expressing +25 GFP. We chose cells of comparable size so that the diffusion rate can be compared visually. (b) The GFP diffusion coefficient plotted against its net charge in all three organisms. The points indicate medians and the error bars show the interquartile range. *P*-values are reported in *Supplementary file 1C*. (c) Histograms of GFP diffusion coefficients for the −7 and +25 variants in all three organisms.

DOI: https://doi.org/10.7554/eLife.30084.004

The following figure supplement is available for figure 2:

**Figure supplement 1.** Unequal fluorescence distribution for +25 GFP in *E. coli* and *Hfx. volcanii*.
DOI: https://doi.org/10.7554/eLife.30084.005

presence of more negative proteins than in *E. coli* and *L. lactis*, which may titrate GFP away from its slower binding partner (see proteome analysis below). We also note that in *Hfx. volcanii* the diffusion coefficient of −30 GFP is higher than of −7 GFP, 10 µm²/s compared to 5.5 µm²/s, which may be caused by a less negative binding partner for GFP in *Hfx. volcanii* than in *E. coli*.

## The effect of osmotic upshift on protein diffusion in *E. coli*

Next, we determined the diffusion coefficient of −30, −7, +15 and +25 GFP in *E. coli* after resuspending the cells in medium with a higher osmolality. It is known that the GFP diffusion coefficient drops drastically after an osmotic upshift (*Konopka et al., 2009*; *Mika et al., 2014*; *Mika et al., 2010*). We now observe what happens after combining two causes for slowed diffusion: osmotic upshift (increased crowding) and protein surface charge. The cells were grown at an osmolality of 0.28 Osm and resuspended in media of 0.55 or 1.2 Osm (adjusted with NaCl). In 0.55 Osm medium the diffusion coefficients did not change much (*Figure 3a* and *Supplementary file 1B*), similar to what was observed before for wildtype GFP (*Konopka et al., 2009*). In 1.2 Osm medium the diffusion coefficients of all variants dropped (*Figure 3a* and *Supplementary file 1B*). The degree of the drop is 56-, 28-, 16-, and 7-fold (between medians) for −30, −7, +15 and +25, respectively, and this difference may be a consequence of the increased cytoplasmic ion concentration that accompanies the osmotic upshift. Thus, the fold-change is less for the positive proteins because the electrostatic screening may compensate partly for the increased crowding effect.

## The distribution of surface charge affects the diffusion coefficient in *E. coli*

For −30, −7, 0, +7, +11b, +15 and +25 GFP, the charge is distributed more or less equally over the surface of the protein. We also determined how a more localized charge affects the diffusion. The +11a variant of GFP has the positive charge unequally distributed, compared to +11b GFP (*Figure 3d*). We determined diffusion coefficients for the +11a and +11b variants by FRAP. Histograms of the diffusion coefficients over populations of cells are shown in *Figure 3c*. The mean diffusion coefficient of +11b GFP is 2.7 µm²/s and that of +11a GFP is 0.76 µm²/s (*Supplementary file 1B*). So, it clearly matters how the net charge is distributed over the GFP surface. For +11a GFP we also see heterogeneous distributions, similar to +15 and +25 GFP, which could be prevented by inducing for a shorter amount of time.

## The +25 GFP variant does not co-localize with DNA

To find out whether the positive GFP variants are bound to DNA or ribosomes-mRNA or both, we first determined the co-localization between GFP and DNA. We expressed +25 GFP in *E. coli*, labelled the nucleoid with DRAQ-5 and shrunk the nucleoid with chloramphenicol. We compared the fluorescence profile, along the length of the cells, of +25 GFP with that of DRAQ-5 (*Figure 4a* and *Figure 4—figure supplement 1*). In all cells the distribution of +25 GFP matched the dimensions of the cells. In nine cells (*Figure 4—figure supplement 1a–i*) out of 46 the nucleoid had shrunk and in all these cells + 25 GFP did not co-localize with DNA. In some cells + 25 GFP was occluded from the DNA, which has been seen before for ribosomes (*Bakshi et al., 2012*). In the other cells the nucleoid did not shrink and the DNA and +25 GFP overlapped (see *Figure 4—figure supplement 1j–p*). We conclude that DNA is not the major binding partner for +25 GFP.

## DNA is not needed for the decrease of +25 GFP diffusion rate

Next, we determined if DNA affects the mobility of +25 GFP by analyzing the diffusion of −7 and +25 GFP in DNA-free regions of *E. coli* cells. We created DNA free regions large enough for FRAP measurements by first treating cells with cephalexin, to elongate the cells, and then with chloramphenicol, to shrink the nucleoid. We visualize the position of DNA by adding DRAQ-5. Only a fraction of the cells had enough GFP fluorescence for FRAP and sufficient DRAQ-5 fluorescence for visualizing the position of DNA, and these were analyzed (*Figure 4—figure supplement 2a and b*). We find that for both −7 and +25 GFP the diffusion coefficient has dropped after treatment of the cells with cephalexin and chloramphenicol (*Figure 4—figure supplement 2c and d*). There is a big difference between the mean diffusion coefficient of −7 and +25 GFP after treatment and in the absence of DNA (35-fold). A similar difference between −7 and +25 GFP was found in cells that

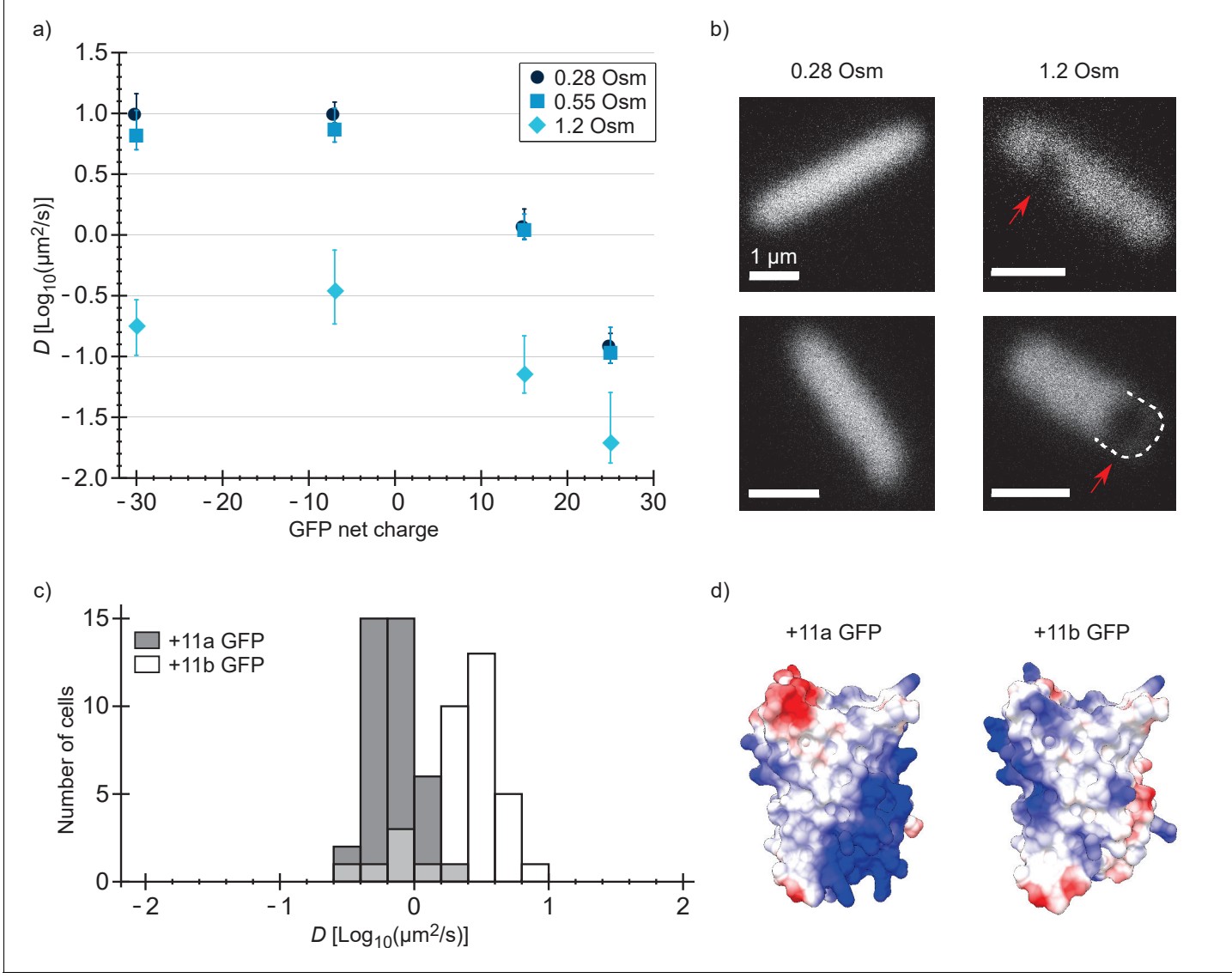

**Figure 3.** Diffusion coefficients of surface-modified variants of GFP at different osmotic stress and charge distribution effects. (a) GFP diffusion coefficients as a function of their net charge and degree of osmotic stress in *E. coli*. The points indicate medians and the error bars show the interquartile range. Discs: *E. coli* cells resuspended in medium with the same osmolality as the growth medium (0.28 Osm); data from *Figure 2b*. Squares: cells resuspended in 0.55 Osm; Diamonds: cells resuspended in 1.2 Osm. *P*-values are reported in *Supplementary file 1C*. (b) Microscopy images of −7 GFP fluorescence of cells resuspended in 0.28 Osm (left panel) and 1.2 Osm medium osmolality (right panel). Red arrows indicate invaginations which appear after rapid osmotic upshift. Scale bars are 1 μm. (c) Histogram of diffusion coefficients for +11a (grey bars) and +11b GFP (white bars) in *E. coli*, measured at growth osmolality (0.28 Osm). The FRAP data only includes cells with homogeneous fluorescence. *P*-values are reported in *Supplementary file 1C*. (d) Structural models of +11a and +11b GFP variants. The colors indicate the charge distribution; the same protein faces are shown.

DOI: https://doi.org/10.7554/eLife.30084.006

were not treated and in the presence of DNA (85-fold). This shows that the drop in +25 GFP diffusion rate is not dependent on DNA. Both −7 and +25 GFP are somewhat excluded from DNA (nucleoid) but the effect is largest for +25 GFP; this is another piece of evidence suggesting that +25 GFP binds to ribosome-mRNA and not to DNA.

## +25 GFP binds ribosome-mRNA

It has been shown that the 30S ribosomal subunit in *E. coli* increases its diffusion coefficient, from 0.04 μm²/s to 0.6 μm²/s, after treatment with rifampicin (*Bakshi et al., 2012*). Rifampicin stops

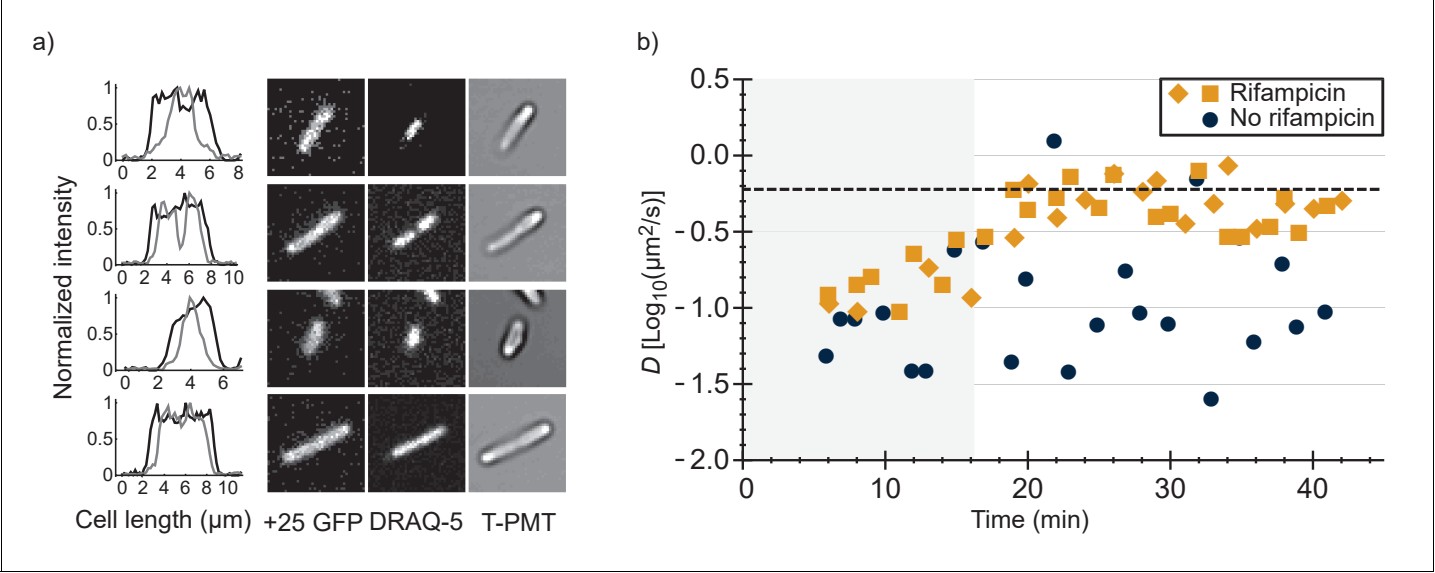

**Figure 4.** Comparison of distributions of +25 GFP and DNA in *E. coli* and diffusion of +25 GFP in the presence and absence of mRNA. (**a**) Co-localization of +25 GFP and DNA in *E. coli*. The plots indicate the fluorescence profile for +25 GFP (black) and DNA (grey) along the length of the cell, averaged over a 5 pixel wide band. The images show the corresponding cells in the +25 GFP, DRAQ-5 (DNA) and T-PMT channels. The T-PMT image corresponds to the transmitted excitation light during the recording of the fluorescence (it is equivalent to a bright-field image). (**b**) Diffusion of +25 GFP in *E. coli* in the presence and absence of mRNA. At time point zero, DMSO + rifampicin (yellow) or DMSO only (blue) was added to the cells. The squares and diamonds indicate different replicates. The dashed line indicates the diffusion coefficient of the 30S ribosome after the addition of rifampicin. At the transition from the shaded region to the white region, >75% of the mRNA is gone.

DOI: https://doi.org/10.7554/eLife.30084.007

The following figure supplements are available for figure 4:

**Figure supplement 1.** Comparison of distributions of +25 GFP and DNA in *E. coli*.
DOI: https://doi.org/10.7554/eLife.30084.008

**Figure supplement 2.** Diffusion coefficients of −7 and +25 GFP in DNA-containing and DNA-free regions.
DOI: https://doi.org/10.7554/eLife.30084.009

transcription and after adding it to *E. coli* cells the pool of mRNA plummets, with 90% of the mRNAs having a half time of less than 8 min (*Bernstein et al., 2002*). We determined the diffusion coefficient of +25 GFP in *E. coli* as a function of time after the addition of rifampicin and compared this to the situation without rifampicin (*Figure 4b*).

We found that after treatment with rifampicin, the diffusion coefficient of +25 GFP increases for 20 min and then levels off. This coincides with the time needed to degrade most of the mRNA. Importantly, the diffusion coefficient after 20 min of rifampicin treatment is close to the value of the 30S and presumably the 50S subunit. We also find that the fluorescence of +25 GFP expressing cells becomes more homogenous after rifampicin treatment. Together, these findings indicate that the positive GFP variants bind to ribosomes, and that this is the major cause for their slow diffusion. We can't rule out that some +25 GFP binds to mRNA, but it is not a major contributor to the decrease of +25 GFP diffusion coefficient. With reasonable confidence we also put aside two other hypotheses: (i) differential partitioning of negative and positive GFPs in different cytoplasmic phases; and (ii) formation of big clusters of positive GFPs with negative proteins. Finally, we find that the variation in diffusion coefficient between cells is smaller in the presence than in the absence of rifampicin, suggesting that part of the spread in the diffusion relates to ongoing transcription.

## Co-localization on sucrose gradients shows that +25 GFP binds predominantly to ribosomes

To substantiate our *in vivo* findings on the binding of positive GFPs to ribosomes, we determined whether −7 and +25 GFP co-localize with ribosomes and/or DNA on a sucrose gradient. For this experiment we used (ribosome containing) lysates of −7 or +25 GFP expressing *E. coli* cells. The cell

lysates were 200–300 times diluted relative to the cytoplasmic contents. The contents of the lysates were separated by centrifugation on a linear sucrose gradient and we determined the presence of the GFPs, by fluorescence spectroscopy, in fractions taken along the length of the gradient. We also determined the presence of ribosomes by electron microscopy. We observed a clear difference in the position of −7 and +25 GFP along the gradient, with the peak of the +25 GFP distribution coinciding with the presence of ribosomes (*Figure 5a and c*). We do not know the exact DNA content of the lysates, so we performed two more experiments in which we added pure DNA to the +25 GFP cell lysates before separating their contents on sucrose gradients. We used a DNA/ribosome ratio that is comparable (0.02 g/L DNA, experiment 1), or five times higher to that in the cell (0.12 g/L DNA, experiment 2) (*Milo and Phillips, 2015*). The position of DNA along the gradient was determined in a separate experiment in which only the pure DNA was added to a sucrose gradient. Even at the highest concentration of DNA, +25 GFP co-localizes with ribosomes and not with DNA (*Figure 5b*). From the combination of sucrose gradient experiments and our *in vivo* studies described above, we conclude that +25 GFP binds predominantly to ribosomes.

## Purified +25 GFP associates and aggregates with purified ribosomes

We also determined if purified ribosomes are able to bind purified +25 or −7 GFP by analyzing solutions of ribosomes mixed with GFPs with size-exclusion chromatography (SEC). At physiological salt concentrations (*Shabala et al., 2009*) we observe that upon mixing ribosomes and +25 GFP, aggregates are formed (*Figure 5e*). These aggregates are not seen when −7 GFP is mixed with ribosomes. In the SEC experiment, the association of ribosomes and +25 GFP results in a decrease of the 280 nm absorbance peak at around 10 mL, because the aggregates are removed by centrifugation (*Figure 5d*). This effect scales with increasing +25 GFP concentration. In line with these observations on binding of +25 GFP to ribosomes, we observe an increase in the fluorescence at the ribosome elution volume when we mix ribosomes and +25 GFP (*Figure 5f*). The control experiment with −7 GFP shows that the anionic protein does not bind to ribosomes and elutes on SEC at around 17.5 ml (*Figure 5g*). From these results we conclude that +25 GFP indeed associates with ribosomes *in vitro*.

## Discussion

### The fraction of GFP variants bound to ribosomes in *E. coli*, *L. lactis* and *Hfx. volcanii*

The diffusion coefficient of GFP in cells is a function of free and bound GFP. If the exchange between these two states is longer than the time of the FRAP measurement one will observe two populations. If the two states exchange on a timescale much shorter than the time of the FRAP measurement the diffusion can be described by a single diffusion coefficient ($D_{eff}$):

$$D_{eff} = f_{free} \, D_{free} + \left(1 - f_{free}\right) D_{bound} \tag{1}$$

We used *Equation (1)* to calculate the fraction of free GFP for each of the GFP variants in *E. coli*, *L. lactis* and *Hfx. volcanii* (see Materials and methods for derivation of *Equation (1)*). We made a number of assumptions: (1) the exchange between free and bound state is much faster than the FRAP measurement; (Vijayakumar et al., 1998) the highest diffusion coefficient of all variants in a given organism reflects the free state of GFP; (3) GFPs bind solely to ribosomes; (4) the total number of binding sites on all ribosomes is higher than the number of GFPs; (Crowley et al., 2011) the decrease of diffusion coefficient with net positive charge has the same origin in all three organisms; and, finally (6) the ribosome diffusion coefficient is the same in all three organisms. Justifications for these assumptions are described in the Materials and methods section. The results are shown in *Figure 6—figure supplement 1*. From the analysis we conclude that even in *Hfx. volcanii*, with its high internal ion concentration, a major fraction (0.81) of +25 GFP is still bound to ribosomes.

Next, we estimated the dissociation constant ($K_d$) of the association between GFP and ribosomes. For *E. coli*, under our growth conditions, the number of ribosomes per $\mu m^3$ is about 17000 (*Vendeville et al., 2011*), which corresponds to a concentration of 10 $\mu M$. GFP probably binds to the RNA that is exposed on the surface of the ribosome and probably does so nonspecifically. The ribosome has a diameter of 20 nm, which yields a surface area of 1260 $nm^2$, assuming a spherical

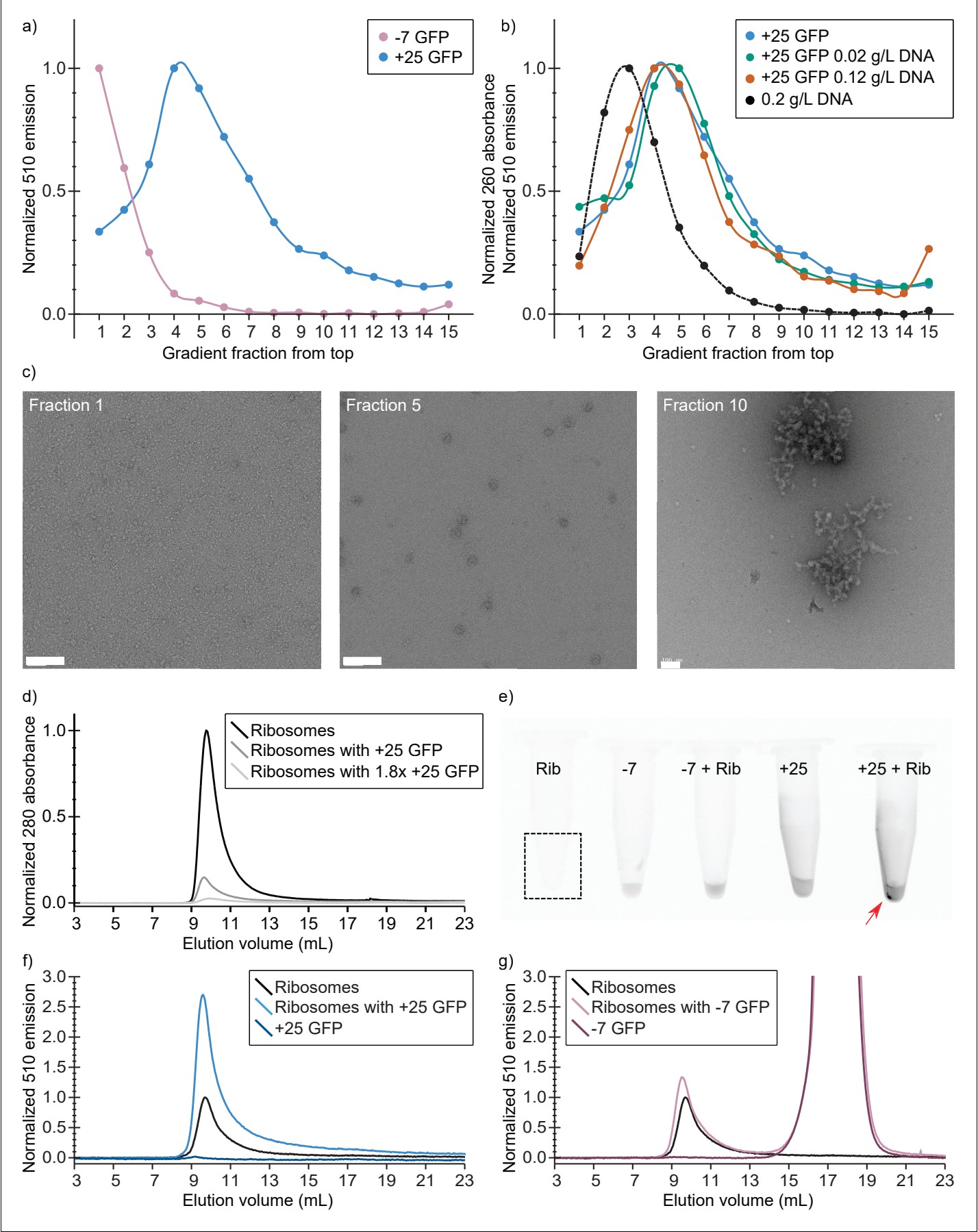

**Figure 5.** *In vitro* experiments show that +25 GFP associates with ribosomes. (**a**) Comparison of fluorescence profiles of sucrose gradient centrifugation experiments performed on *E. coli* cell lysates containing either −7 or +25 GFP. The majority of −7 GFP is present in the loaded sample (fractions 1 and 2), while +25 GFP peaks at fractions 4–5, corresponding to 15–18% (w/v) sucrose. The fluorescence signals were normalized to the highest value, because the absolute fluorescence of +25 GFP is lower than that of −7 GFP. (**b**) Comparison of fluorescence (and absorption) profiles of sucrose gradient centrifugation experiments performed on purified DNA (0.2 g/L), and *E. coli* cell lysates containing +25 GFP with or without additional DNA. (**c**) Transmission electron microscopy images of uranyl acetate-stained fractions from the cell lysate containing sucrose gradients. Fraction one lacks distinct large structures, whereas fraction 10 shows large aggregates. Ribosomes, spheres of around 25–30 nm diameter, are visible and peak in fraction 5. The scale bar is 100 nm. (**d**) Elution profiles on a Sephadex 200 of pure ribosomes, ribosomes with +25 GFP, and ribosomes with 1.8x as much +25 GFP, measured as absorbance at 280 nm (**e**) Fluorescence imaging of tubes containing ribosomes, −7 GFP, ribosomes mixed with −7 GFP, +25 GFP, and ribosomes mixed with +25 GFP. The indicated fluorescent pellet appears after centrifuging the samples. The tube containing only ribosomes is not visible due to lack of fluorescence. (**f**) Elution profiles of ribosomes, +25 GFP, and ribosomes mixed with +25 GFP, measured as fluorescence at 510 nm; the excitation wavelength was 488 nm. The higher fluorescence of ribosomes mixed with +25 GFP is accompanied by decrease in absorbance at 280 nm; (**g**) Elution profiles of ribosomes, −7 GFP, and ribosomes mixed with −7 GFP, measured as fluorescence at 510 nm; the excitation wavelength was 488 nm. Unlike for +25 GFP, when ribosomes are mixed with −7 GFP the absorbance at 280 nm remains the same (data not shown).

DOI: https://doi.org/10.7554/eLife.30084.010

shape. About half of this surface area is RNA so we end up with 630 nm$^2$. The diameter of GFP is 3.5 nm, giving a 9.6 nm$^2$ cross section. Dividing the ribosome RNA surface area by the GFP cross section gives a maximal number of binding sites of 66. This means that the concentration of binding sites is 660 µM. To calculate $K_d$ we use:

$$f_{bound} = \frac{[binding\ site]}{K_d + [binding\ site]} \tag{2}$$

This equation is valid when the number of binding sites is significantly higher than the number of GFPs. Using a fraction of bound +25 GFP of 0.99 in *E. coli*, we obtain $K_d$ = 6.7 µM. If we make the assumption that the concentration of ribosomes in *L. lactis* and *Hfx. volcanii* is the same as in *E. coli*, then the $K_d$ for binding of +25 GFP to a ribosome binding site is 65 µM for *L. lactis* and 155 µM for *Hfx. volcanii*.

## The relation between diffusion coefficient, GFP net charge and ionic strength

In this section we seek to explain: (i) the relation between $D_{eff}$ and cytoplasmic ionic strength; and (ii) the relation between $D_{eff}$ and net charge of GFP. To explain (i) we first compare our values to literature data on electrostatic interactions between proteins in dilute solution. To make the comparison possible we relate $K_d$, rather than $D_{eff}$, to ionic strength. In *Figure 6a* we plot the $K_d$ *versus* ionic strength of the interaction between barnase and barstar (*Schreiber and Fersht, 1993*), colicinE9 and Im9 (*Wallis et al., 1995*), and different forms of thrombin and hirudin (*Stone et al., 1989*).

The ionic strength dependence of the interaction of +25 GFP with ribosomes (recorded in *E. coli*, *L. lactis* and *Hfx. volcanii*; *Figure 6b*) is similar to that of the three protein pairs in dilute solutions (*Figure 6a*); the $K_d$ increases with ionic strength and levels off at higher ionic strength. To get more insight into the interaction we applied a semi-empirical equation that was successfully used to describe the (partly) electrostatic interaction between the proteins thrombin and hirudin (*Stone et al., 1989*):

$$\Delta G_b^\circ = \Delta G_{nio}^\circ + \Delta G_{io0}^\circ \frac{e^{-C_1\sqrt{I}}}{1 + C_1\sqrt{I}} \tag{3}$$

This equation was derived from Debye-Hückel theory and was subsequently modified to account for behavior at higher ionic strength. Here $\Delta G_b^\circ$ is the total binding energy, $\Delta G_{nio}^\circ$ is the binding energy due to non-ionic interactions, $\Delta G_{io0}^\circ$ is the binding energy due to electrostatic interactions in the absence of ions, $C_1$ is a parameter that depends on the distance between charges and the screening effects that are not due to ions, and $I$ is the ionic strength. To apply *Equation 3* we need to relate the $K_D$ and $\Delta G_b^\circ$:

$$K_D = e^{\frac{\Delta G_b^\circ}{RT}} \tag{4}$$

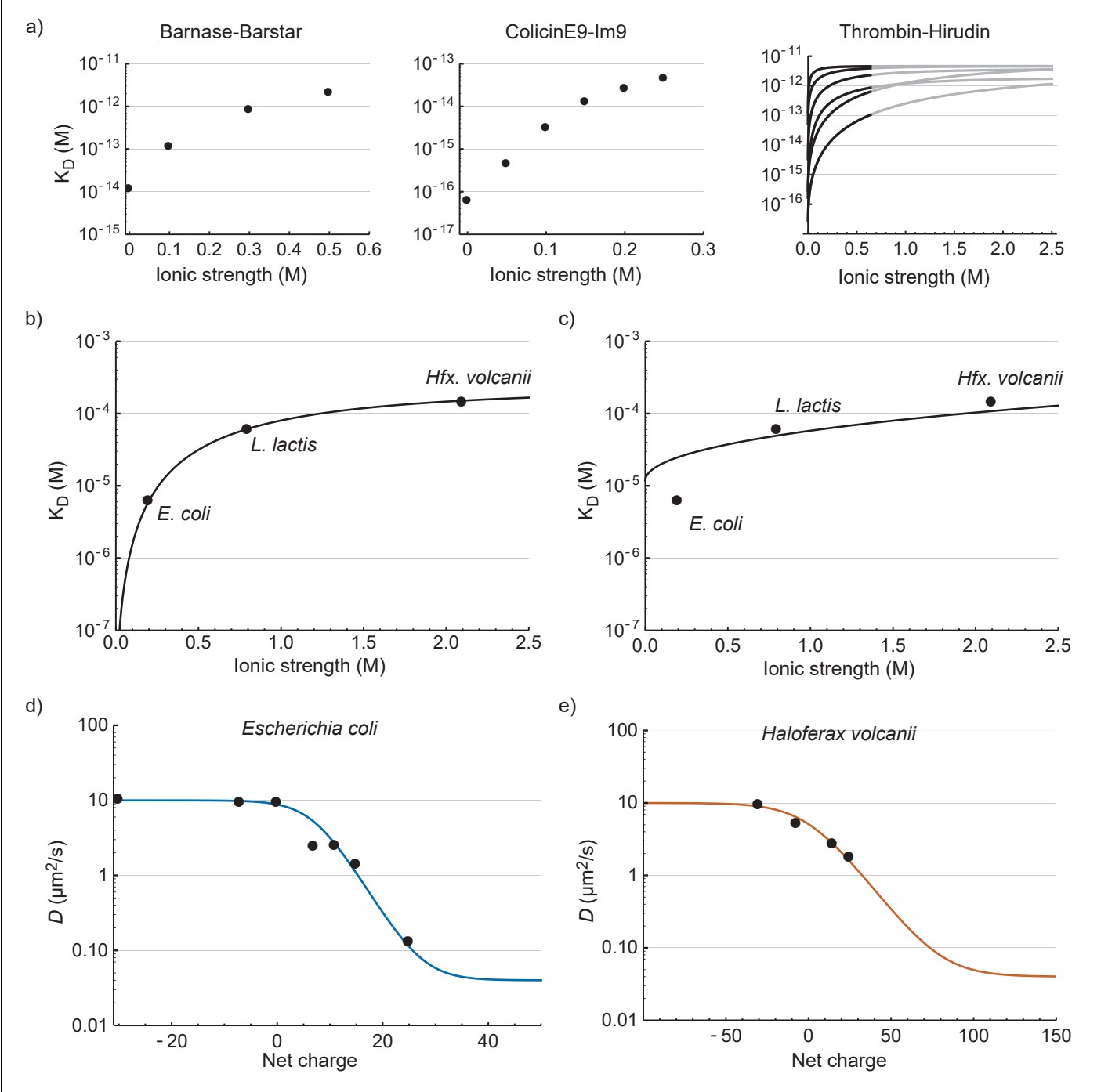

**Figure 6.** The relation between the diffusion coefficient, net charge and ionic strength. (**a**) The dependence of dissociation coefficients on ionic strength for protein binding pairs barnase-barstar, colicinE9-Im9, and different variants of hirudin binding to thrombin. The data from the literature was in the form of $k_{on}$ and $k_{off}$ and we used $K_D = \frac{k_{off}}{k_{on}}$ to determine the affinity constants. For the thrombin-hirudin interactions we show fits with a combination of *Equations (3) and (4)*. The black part spans the data, the grey part is an extrapolation. The charge on hirudin decreases from the bottom to the top line (at the black part). (**b**) The dependence of dissociation coefficients on ionic strength for +25 GFP. The black line is a fit with a combination of *Equations (3) and (4)*. (**c**) Same data as in (**b**) but with the non-ionic contribution to the binding free energy fixed at zero during fitting. (**d**) The dependence of diffusion coefficient on GFP net charge for *E. coli*. The line is a fit with *Equation 5*. We did not include +11a GFP because of its different charge distribution. (**e**) The dependence of diffusion coefficient on GFP net charge for *Hfx. volcanii*. The line is a fit with *Equation 5*.
DOI: https://doi.org/10.7554/eLife.30084.011

The following source data and figure supplements are available for figure 6:

*Figure 6 continued on next page*

*Figure 6 continued*

**Figure supplement 1.** Fraction of free GFP variants in *E. coli*, *L. lactis* and *Hfx. volcanii*.
DOI: https://doi.org/10.7554/eLife.30084.012
**Figure supplement 1—source data 1.** Free fraction of GFP variants in *E. coli*, *L. lactis* and *Hfx. volcanii*.
DOI: https://doi.org/10.7554/eLife.30084.013

We fitted the +25 GFP interaction data with a combination of *Equations (3) and (4)* and obtained the following parameter values: $\Delta G^{\circ}_{nio}$ = −20 400 J mol$^{-1}$, $\Delta G^{\circ}_{io0}$ = −28 900 J mol$^{-1}$, and $C_1$ = 1.53. We had expected that the non-ionic interaction free energy would be close to zero, but if we impose this conditions the fit becomes bad (*Figure 6c*). The bending off at higher ionic strengths depends on $\Delta G^{\circ}_{io0}$ being negative; we expect $\Delta G^{\circ}_{io0}$ to be negative given the electrostatic attraction between positive GFP and negatively-charged surfaces of the ribosomes.

The second phenomenon we explain is the relation between $D_{eff}$ and the surface charge of GFP. In the work describing the hirudin-thrombin interaction a number of charge variants of hirudin were used (*Stone et al., 1989*). The $\Delta G^{\circ}_{io0}$ depended linearly on the number of charges. This is expected from Coulombs law, assuming that non-linearities do not arise from charge screening. In the rest of the analysis we assume that $\Delta G^{\circ}_{io0}$ indeed depends linearly on the charge and we write: $\Delta G^{\circ}_{io0} = \Delta G^{\circ}_{pc} \times charge$, where $\Delta G^{\circ}_{pc}$ is the free energy change per charge. We can now combine this with *Equations (1), (2), (3), and (4)* to obtain:

$$D_{eff} = \left(1 - \frac{[binding\ site]}{[binding\ site] + e^{\frac{\Delta G^{\circ}_{nio} + \Delta G^{\circ}_{pc}\ charge \frac{\exp(-C_1\sqrt{I})}{1+C_1\sqrt{I}}}{RT}}}\right)(D_{free} - D_{bound}) + D_{bound} \qquad (5)$$

We used this equation to fit the data for *E. coli* and *Hfx. volcanii*. We set $D_{free}$ = 10 μm$^2$/s, $D_{bound}$ = 0.04 μm$^2$/s (ribosome diffusion coefficient), $[binding\ site]$ = 660 μM, $T$ = 293 K, $R$ = 8.314 J K$^{-1}$ mol$^{-1}$ (gas constant), and $I$ = 0.2 M for *E. coli* and $I$ = 2.1 M for *Hfx. volcanii*. We are left with three fitting parameters: $\Delta G^{\circ}_{nio}$, $\Delta G^{\circ}_{pc}$, and $C_1$. The fits are shown in *Figure 6d and e* (see *Supplementary file 1D* for fitting parameters). The model fits the data well; this is more telling for *E. coli* than it is for *Hfx. volcanii*, as the data covers more of the curve. The upper bound for the diffusion coefficient is set by free diffusion and the lower bound by the diffusion of the ribosome. When we use a log scale to represent the diffusion coefficient we see a linear dependence of diffusion coefficient with net charge from 0 to +25 (for *E. coli*). Thus, under the assumption that $\Delta G^{\circ}_{io0}$ depends linearly on the number of charges the model reproduces the linear dependence between the upper and lower bound. Again, we needed to include a non-ionic interaction for a proper fit. There is a discrepancy between the parameter values of the fits shown in *Figure 6b,d and e* (see *Supplementary file 1D*). This may be caused by the assumptions made above not holding up. Together the results for the relation between $K_d$ and ionic strength, and $D_{eff}$ and GFP charge, show that the binding of GFP to ribosomes can be described by electrostatic interactions and screening by small ions on top of a base non-ionic interaction component.

## Proteome analysis reveals potentially slow proteins

To determine the consequences of our findings we analyzed the proteomes of *E. coli*, *L. lactis* and *Hfx. volcanii* and four (endo)symbiotic bacteria. We determine (i) how a protein will diffuse in light of the composition of the proteome, for which we need to know both the net charges and the abundances of all proteome components; and (ii) how the ribosomes affect the diffusion of the proteome constituents, for which we need to know the net charge of the proteins. We determined the distributions of pI values and net charges for all proteins in the genome and for only cytoplasmic proteins, in *E. coli*, *L. lactis* and *Hfx. volcanii* (*Figure 7a*). We also determined the distribution of pI values and net charges for cytoplasmic proteins in *E. coli* taking into account protein copy numbers (*Figure 7—figure supplement 1*).

From the pI distributions it is clear that in all three organisms the majority of cytoplasmic proteins is acidic and thus negatively charged at the physiological internal pH of 7.5; for internal pH values

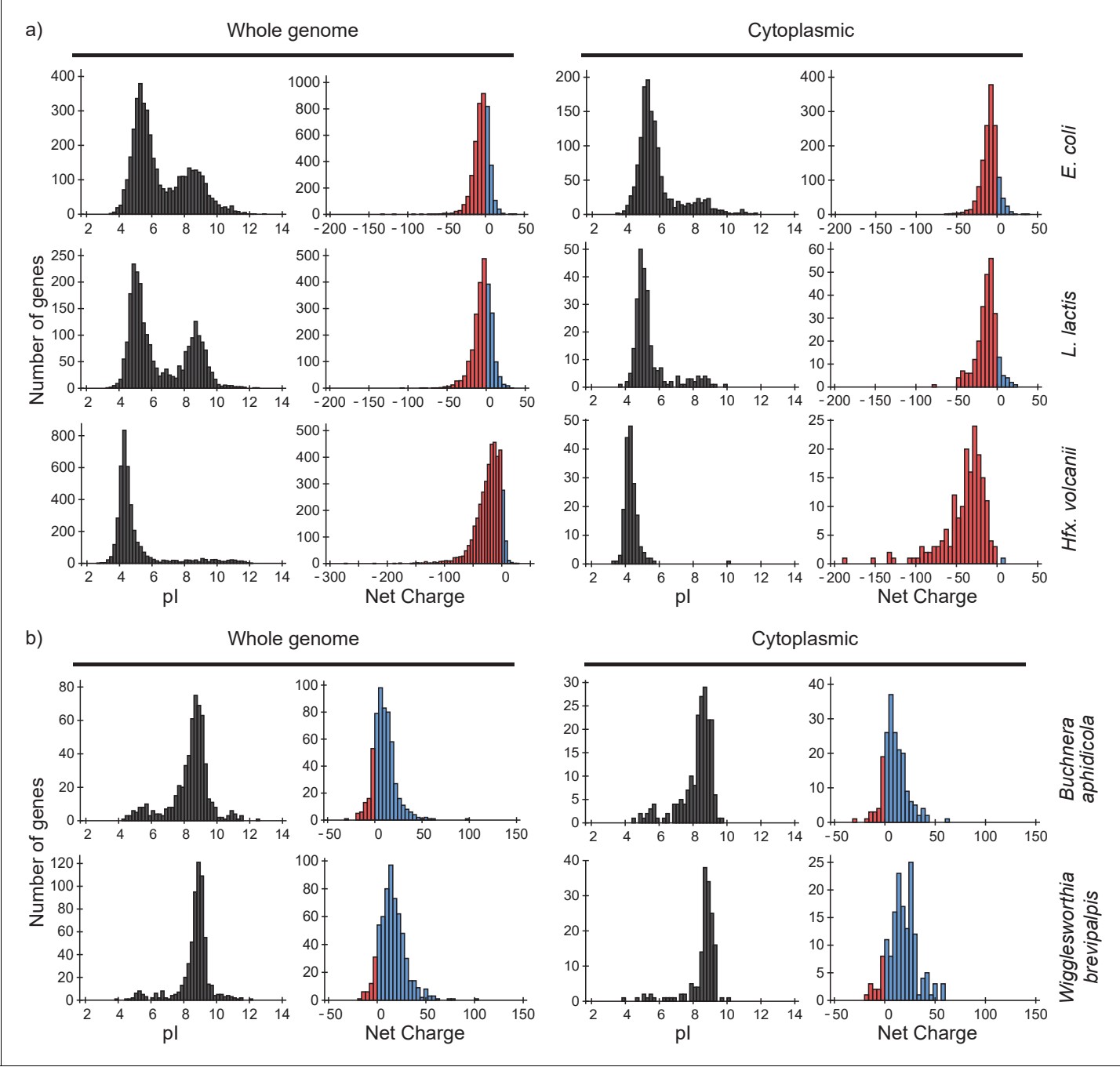

**Figure 7.** pI and net charge distributions for proteins of *E. coli*, *L. lactis*, *Hfx. volcanii*, *Buchnera aphidicola* and *Wiggleworthia glossinidia brevipalpis*. The histograms show the number of genes that encode proteins with given pI and net charge. We show distributions over all genes (left panels) and over genes that encode cytoplasmically localized proteins (right panels). (a) *E. coli*, *L. lactis*, and *Hfx. volcanii*; (b) the two symbionts, *Buchnera aphidicola* and *Wigglesworthia glossinidia brevipalpis*, that have the most positive proteomes (from the four symbionts that we analysed). We used gene ontology annotations from the UniProt database to find the cytoplasmic proteins. In all cases we assumed a pH of 7.5 for calculating the net charge. The symbionts were selected based on pI profiles from (*Kiraga et al., 2007*).

DOI: https://doi.org/10.7554/eLife.30084.014

The following figure supplements are available for figure 7:

**Figure supplement 1.** pI and net charge distributions for the *E. coli* proteome, taking into account protein abundance.

DOI: https://doi.org/10.7554/eLife.30084.015

**Figure supplement 2.** Protein pI and net charge distributions for *Buchnera aphidicola*, *Blochmannia floridanus*, *Onion yellows phytoplasma*, and *Wigglesworthia glossinidia brevipalpis*.

*Figure 7 continued on next page*

*Figure 7 continued*

DOI: https://doi.org/10.7554/eLife.30084.016

we refer to (*Slonczewski et al., 1981*; *Zilberstein et al., 1982*; *Wilks and Slonczewski, 2007*) for *E. coli* (see also BNID 105980 and BNID 106518) (*Milo et al., 2010*), and (*Poolman et al., 1987*) for *L. lactis*. The protein net charge distributions of *E. coli* and *L. lactis* go up to a value of +25, irrespective of whether we take the gene-based distributions or protein copy numbers (for *E. coli*), or whether we take the full or cytoplasmic proteome. The net charge distribution of *Hfx. volcanii* stops at about 0. In *E. coli* 35 cytoplasmic proteins have a net charge higher than +10. These consist of 18 ribosomal, 9 RNA-associated, 5 DNA-associated and 3 uncharacterized proteins. For *L. lactis*, with seven cytoplasmic proteins that have a net charge bigger than +10, the breakdown is similar. The only *Hfx. volcanii* protein with a net charge bigger than 0 is a ribonuclease, rnp4, with a net charge of +6. Thus, all three organisms essentially have no 'free' positive cytoplasmic proteins.

The drop in diffusion coefficient with increased positive charge may partly explain why the proteome is mostly negative in *E. coli*, *L. lactis* and *Hfx. volcanii*. Positive proteins not only diffuse slowly but their binding to ribosomes might also inhibit protein synthesis, that is, by affecting the assembly or the activity of ribosomes. As mentioned, there are 18 ribosomal proteins in *E. coli* that have a net charge of more than +10. If the findings on GFP are transferable to ribosomal proteins, then these ribosomal proteins experience a drop in diffusion coefficient of more than 5-fold. The most extreme cases, RplT and RplB, with net charges of +24 and +31, would have a drop in diffusion coefficient of 100-fold. The fact that ribosomal proteins themselves are positively charged potentially limits the rate of assembly of new ribosomes. The reason is that ribosomal proteins that have just been synthesized bind nonspecifically to the surfaces of fully assembled ribosomes, and this causes slow diffusion, lower effective protein concentrations and may affect the functioning of ribosomes. An implication of our findings is that the synthesis of cationic ribosomal proteins and the assembly of ribosomes should be highly coordinated and preferably be modular to minimize unwanted side-effects of nonspecific interactions (*Davis et al., 2016*).

A high positive net charge is not a guarantee for binding to the ribosome. One could imagine a positive protein that is disordered before binding but ordered upon binding. The reduction in entropy reduces the binding affinity, which in turn causes the diffusion coefficient to be high. Another option is that the surface shapes don't match even if the net charges are complementary. This would also lower the affinity and increase the diffusion coefficient.

A conundrum is encountered when we look at the proteomes of the bacteria *Buchnera aphidicola*, *Blochmannia floridanus*, *Onion yellows phytoplasma*, and *Wigglesworthia glossinidia brevipalpis*. All four organisms are (endo)symbionts of plants or insects and have small genomes: 572–730 protein-encoding genes. All of these have very basic proteomes (*Figure 7b* and *Figure 7—figure supplement 2*). It is unclear how these organisms are able to deal with, or avoid, slow diffusion and ribosomes getting swarmed with positive proteins.

## A few general points on diffusion and binding

A protein can diffuse only as slow as the combination of this protein and the slowest component it binds to in the cell. This is described by *Equation (1)*, which can be rewritten in the following way:

$$D_{eff} = f_{free} \left( D_{free} - D_{bound} \right) + D_{bound} \tag{6}$$

If we fill in $f_{free} = 0$ we get $D_{eff} = D_{bound}$ and if we fill in $f_{free} = 1$ we get $D_{eff} = D_{free}$; the black lines in *Figure 8*. If we fill in $D_{bound} = 0$ we get $D_{eff} = f_{free} D_{free}$, i.e. after some point $D_{eff}$ becomes independent of $D_{bound}$; this is illustrated by the grey lines in *Figure 8*. In more concrete terms: 99% binding of +25 GFP to ribosomes ($D_{ribo}$=0.04 μm$^2$/s) leads to the same diffusion coefficient as 99% binding to DNA ($D_{DNA}$=0.000035–0.00007 μm$^2$/s).

## Conclusion

We find that the diffusion coefficients of proteins in the cytoplasm of *E. coli* depend on their net charge and the distribution of charge over the protein surface, with positive proteins moving up to 100-fold slower. The diffusion becomes even slower when cells are exposed to an osmotic upshift. In

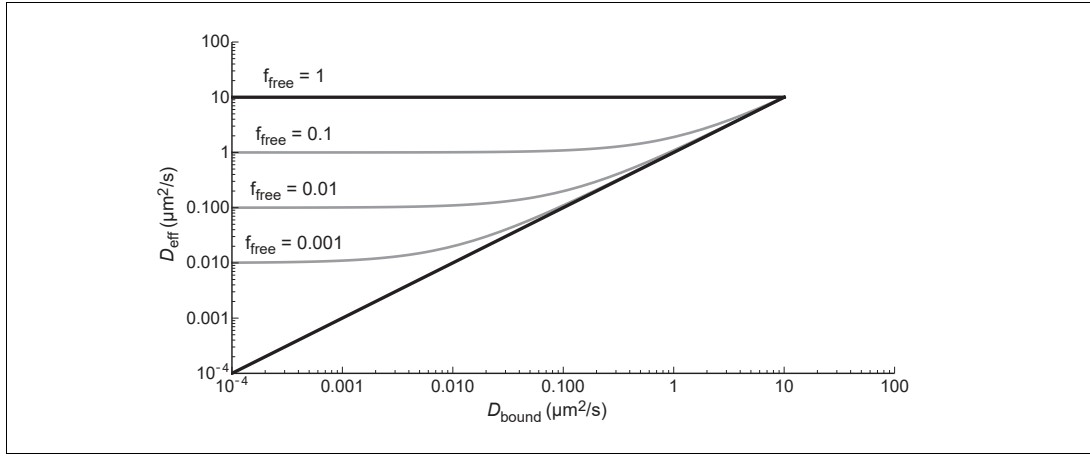

**Figure 8.** The effective diffusion coefficient as a function of free fraction and the diffusion coefficient of the bound complex. All lines are generated with **Equation (6)** and $D_{free}$ = 10 $\mu m^2$/s.
DOI: https://doi.org/10.7554/eLife.30084.017

*L. lactis* and *Hfx. volcanii* the slowdown in diffusion with increasing positive surface charge is less than in *E. coli* due to electrostatic screening. The decrease in diffusion coefficient is mainly caused by binding of positive proteins to ribosomes, with $K_D$ values on the order of $\mu M$; this shows that non-selected interactions need not be weak. Ribosome surface properties may thus limit the composition of the cytoplasmic proteome. These findings are of general value due to the universal presence of ribosomes in cells. Application of these findings to bacterial (endo)symbionts lays bare a paradox in the functioning of these cells.

## Materials and methods

### Strains used

We use *E. coli* strain MG1655 (*Blattner et al., 1997*). The GFP variants, −30, –7, 0, +7, +11a, +11b, +15, and +25, were all expressed from an L-arabinose inducible promoter on a pBAD vector with an ampicillin-resistance selection marker. We obtained the genes for −30, –7, +15, and +25 from David Liu's lab at Harvard, and nucleotide sequences of +7, +11a, +11b from David Thompson. See (*McNaughton et al., 2009*) for −30, –7, +15, and +25 GFP; see (*Thompson et al., 2012a*) for +7, +11a and +11b. We designed the 0 GFP ourselves. All GFP variants have an N-terminal histag. For all GFP variants amino acid sequences are available in *Supplementary file 1A*.

We used *Lactococcus lactis* strain NZ9000 (*Linares et al., 2010*), which contains the *nisR* and *nisK* genes which in the presence of the inducer, nisin A, switches on the expression of genes from the *nisA* promoter (*Kuipers et al., 1998*). We cloned the coding segments for the his-tagged versions of −30, −7, +15 and +25 GFP behind the *nisA* promoter in the pNZ8048 vector (*Kuipers et al., 1998*). The expression levels of −30 GFP were too low for FRAP analysis.

We used *Hfx. volcanii* strain H1895 (*Strillinger et al., 2016*). We expressed the GFP variants, −30, –7, +15, and +25, from the pTA1228 plasmid, which has a tryptophan-inducible promoter (*Brendel et al., 2014*). The amino acid sequences are the same as for *E. coli* and *L. lactis*. We optimized the nucleotide sequence for *Hfx. volcanii* by making the codon frequency in the GFP variants the same as for protein coding genes on the *Hfx. volcanii* chromosome. We constructed the pTA1228 bearing the genes for GFP variants in *E. coli*, and transformed the final plasmids in *Hfx. volcanii* by protocols described in the Halohandbook (*Holmes et al., 2008*). We obtained the strain, plasmid and the codon usage table from Thorsten Allers (University of Nottingham, UK).

## Preparation of *E. coli* for FRAP

For each experiment we took a glycerol stock of *E. coli* with one of the GFP variants and stabbed it with a pipette tip to obtain a small amount of cells. These we deposited in 4 mL LB medium containing 0.2 % v/v glycerol and 100 µg/mL ampicillin. We incubated the culture at 30°C, with 200 rpm shaking. The next day we took 8 µL of the LB culture to inoculate 4 mL MBM containing 0.2 % v/v glycerol and 100 µg/mL ampicillin. The composition of MBM, MOPS based medium, can be found in (*Neidhardt et al., 1974*). We adjusted the osmolality of MBM to 0.28 Osm with NaCl. Osmolalities were measured with an Osmomat 030 cryoscopic osmometer (Gonotec, Berlin, Germany). For the −30 GFP variant we also added 0.4% L-arabinose (from a 20 % w/v stock in MilliQ), to induce protein expression. Again we incubated at 30°C, with 200 rpm shaking. The next morning the cultures had reached an $OD_{600}$ between 0.6–1.6, which were diluted to an $OD_{600}$ of 0.18–0.25. At the moment of dilution we also induced the expression of −7 GFP, with 0.1 % w/v L-arabinose, and 0, +7 and +11b GFP, with 0.4 % w/v L-arabinose. We incubated the cultures for a further 2–4 hr, to obtain an $OD_{600}$ of 0.4–0.5, and then performed the FRAP measurements. To avoid aggregation, the +11a, +15 and +25 GFP variants were induced with 0.4 % w/v L-arabinose 1–2 hr before the FRAP measurements. For each GFP variant the experiment was repeated at least three times.

## Preparation of *L. lactis* for FRAP

For each experiment we took a glycerol stock from the −80°C freezer and stabbed it with a pipette tip to obtain a small amount of cells. These cells were deposited in 4 mL of growth medium in a culturing tube. The growth medium was CDM in all cases; the formulation of the chemically defined medium (CDM) is given in the supplement of (*Mika et al., 2014*), where it is referred to as $CDM^{RP}$. There is one difference, here we also added L-proline (0.68 g/L, final concentration). We include glucose (1 % w/v), as a carbon and energy source, and chloramphenicol (5 µg/mL), to maintain the plasmids. We incubated the cultures at 30°C, without shaking (*L. lactis* grows semi-anaerobically). In the morning of the next day about 100 uL of culture was added to 4 mL of fresh CDM, to yield an $OD_{600}$ of about 0.1. Simultaneously, we added 4 uL of nisin A solution (filtered supernatant from a *L. lactis* NZ9700 culture). The cultures were incubated at 30°C. We used the cultures for FRAP measurements at an $OD_{600}$ of 0.38–0.46. We diluted the cultures to keep them from overgrowing. For each GFP variant the experiment was repeated 2–3 times.

## Preparation of *Hfx. volcanii* for FRAP

For each experiment we took 2–3 colonies of *Hfx. volcanii*, expressing −30, –7, +15, or +25 GFP, from an Hv-YPC agarose plate and suspended these in 4 mL Hv-YPC medium (*Holmes et al., 2008*). We incubated the cultures at 42°C, with 200 rpm shaking. The next morning we diluted the culture to an $OD_{600}$ of 0.2–0.3 and at the same time added 4 mM L-tryptophan, to induce expression of the GFP variants. We incubated the cultures for 2–3 hr at 42°C, with 200 rpm shaking, before using the cells for FRAP measurements. At the time of the measurements the $OD_{600}$ was 0.3–0.5. For each GFP variant the experiment was repeated three times.

## Determination of diffusion coefficients

We performed fluorescence recovery after photo-bleaching (FRAP; see *Figure 1a,b*) on a LSM710 Zeiss confocal laser scanning microscope (Zeiss, Oberkochen, Germany), following a method originally described in (*Elowitz et al., 1999*). Our implementation of this method is described in (*Mika et al., 2014*). We started with an overview image containing many cells, from which picked cells that are lying flat, are not undergoing cell division and have no neighbors that would obscure the analysis. We take a high resolution close-up of the cell and its immediate surroundings to see if the cell is fit for measurements. For FRAP we programmed the microscope to take three images, then photo bleach the GFP at one of the cell poles and finally record the recovery of the fluorescence over time. We recorded all images with a 488 nm laser; the same laser was used for bleaching but at a higher power.

For *E. coli* we did the FRAP measurements as follows. We took 400 µL culture and resuspended those cells twice in 300 µL MBM*. MBM* did not have glycerol or ampicillin and has $Na^+$ instead of $K^+$. The osmolality of the MBM* was either 0.28, 0.55 or 1.2 Osm. We adjusted the osmolality of the resuspension medium with NaCl. After we resuspended the cells we pipetted 4 µL of these cells on

a cover slide. To make sure the *E. coli* cells didn't move we used (3-aminopropyl)triethoxysilane (APTES)-treated cover slides. The slides were prepared as follows. First we cleaned them by sonicating for 1 hr in 5 M KOH, rinsing 10 times with MilliQ and blowing off the remaining MilliQ with pressurized nitrogen. We then deposited the slides in acetone that contained 2 % v/v (3-aminopropyl) triethoxysilane. We incubated for 5 min at room temperature, removed the acetone and APTES and rinsed the slides 10 times with MilliQ. Again, the remaining MilliQ was blown off with pressurized nitrogen. After putting our cells on the APTES slide we put an object slide on top, for stability, and put the whole on the microscope stage. The stage temperature was maintained at 30°C. We used the slide for no longer than 20 min after depositing the cells. For −30, –7 and 0 GFP under normal conditions, that is, no osmotic upshift, we recorded images at 50 time points and with 8 × 8 pixels, with a 4 ms exposure time and no extra time between exposures. For the positive GFP variants, and for the FRAP experiments after osmotic upshift (slower diffusion), we recorded images at a 100 time points, with 16 × 16 pixels, and a 8 ms exposure and a 8–100 ms time step between the start subsequent exposures.

For *L. lactis* we pipetted 4 uL culture on a cover slide. We put an object slide on top, for stability, and the whole was put on the stage of the microscope (maintained at 30°C). We made sure that the cells didn't move by using cover slides that were sonicated for 1 hr in 5 M KOH, rinsed 10 times with MilliQ and dried by blowing of the remaining MilliQ with pressurized nitrogen. We used cover slides for no longer than 20 min after depositing cells, and cultures for no longer than 1 hr after reaching an $OD_{600}$ of 0.38–0.46. We recorded the images in 8 × 10 pixels and with an exposure time of 5–37 ms, short for −7 GFP and long for +25 GFP. For each cell we recorded the whole FRAP measurement in 50 images, without extra time between exposures.

For *Hfx. volcanii* we pipetted 4 µL culture on a cover slide (non-treated). Then a patch of 1% agar in 18% SW was put on top, to immobilize the cells (see the Halohandbook (*Holmes et al., 2008*) for the composition of SW). We then put the sample on the microscope stage. The stage temperature was maintained at 30°C. After a couple of minutes the cells stopped moving and we performed our FRAP measurements. We imaged for 20 min on a single sample. We recorded the images in 16 × 16 pixels with an exposure time of 8 ms and a time step, the time between the start of subsequent exposures, of 8–20 ms. For +15 and +25 GFP we see clear aggregates in some cells (*Figure 2—figure supplement 1a*). We did not include these cells in the FRAP measurements.

Overall about 10% of the analyzed cells were too noisy and/or did not show a bleached area, so that a diffusion coefficient could not be determined. Those cells were excluded from the analysis. For the FRAP results of −30 GFP (1.2 Osm shock), −7 GFP and 0 GFP in *E. coli* we excluded around 20% of the cells. For 0 GFP this was necessary because the fluorescence level was too low in a fraction of the cells; for −30 (shock) and −7 GFP the apparent diffusion was too fast for detection. When we include the excluded cells we obtain the following medians: 0.28 $\mu m^2$/s for −30 GFP (shock), 11.3 $\mu m^2$/s for −7 GFP and 9.8 $\mu m^2$/s for 0 GFP. This can be compared to the values 0.18, 10 and 8.6 $\mu m^2$/s reported in *Supplementary file 1B*.

## Co-localization of +25 GFP and DNA in *E. coli*

We grew *E. coli*, containing +25 GFP, in a culturing tube with 4 mL EZ; a rich defined growth medium (Teknova, Hollister, CA, USA), to which we added glycerol (0.2 % v/v), as a carbon and energy source, and ampicillin (100 µg/mL), to maintain the plasmid. We used EZ medium because the chloramphenicol didn't condense the DNA in cells grown in MBM. The culture was incubated at 30°C, with 200 rpm shaking for aeration. The next morning we used 200 uL of this culture to inoculate 4 mL of fresh EZ medium. We incubated the culture for 1 hr before adding 0.1% (w/v) L-arabinose, to induce the expression of +25 GFP. After another hour of incubation, at an $OD_{600}$ of 0.5–0.6, we added DRAQ-5 (2 µM), to visualize the DNA, and chloramphenicol (200 µg/mL), to condense the DNA (*Bakshi et al., 2012*; *Chai et al., 2014*). We imaged these cells between 1 and 1.5 hr after adding the DRAQ-5 and chloramphenicol, for which we deposited 10 uL culture on an APTES cover slide, prepared as described above, put an object slide on top and put the whole on the microscope stage. We performed all measurements at 30°C. We focused on a 200 µm x 200 µm area, containing ~200 cells, and recorded an image in the 488 (+25 GFP) and 633 (DNA) channels and used an exposure time of 3.3 s. We also recorded the transmitted excitation light to obtain a bright-field image. We picked the cells used for analysis from the transmission image to avoid bias. We selected cells that were lying flat and in focus.

## Diffusion coefficients in DNA-free regions

We did FRAP measurements on −7 and +25 GFP in DNA free regions in *E. coli*. The cells were grown in a culturing tube with 4 mL LB medium, containing glycerol (0.2 % v/v), as a carbon and energy source, and ampicillin (100 µg/mL), to maintain the plasmid. We incubated the culture overnight at 37°C, with 200 rpm shaking. We used LB medium, and incubated at 37°C, to get elongated cells in a reasonable time window upon cephalexin treatment (see below). The next day we made two new cultures, 4 mL LB (same composition as above), by adding 4 µL or 16 µL of overnight culture. We incubated these cultures at 37°C, with 200 rpm shaking. At an $OD_{600}$ between 0.19–0.26 we added L-arabinose (0.1–0.2 % w/v), to induce GFP expression, and cephalexin (25 µg/mL), to elongate the cells (*Chai et al., 2014*). After a further two hours of incubation we added chloramphenicol (200 µg/mL), to condense the DNA (*Chai et al., 2014*), and DRAQ-5 (10 µM), to stain the DNA. The concentration of DRAQ-5 is above the minimum inhibitory concentration, 5 µM, for growth of *E. coli* MG1655 in EZ medium (*Bakshi et al., 2012*); at lower concentrations of DRAQ-5 the DNA did not stain properly. After a further 30–70 min incubation we took 200–400 µL of culture and resuspended the culture twice in 200 µL LB (with 0.2 % v/v glycerol, 100 µg/mL ampicillin and 200 µg/mL chloramphenicol), to get rid of the DRAQ-5 background fluorescence. We used the following stock solutions: 20 % w/v L-arabinose in MilliQ, 1 mg/mL cephalexin in MilliQ, 20 mg/mL chloramphenicol in ethanol, and 200 µM DRAQ-5 in MilliQ.

We deposited 4 µL of the sample on an APTES cover slide, put an object slide on top and put the whole on the microscope stage. The microscope stage temperature was maintained at 37°C. The cells were on the stage for no longer than 1 hr. We measured the diffusion coefficients of −7 and +25 GFP in these treated cells by the same method as before. The exception is that we do not draw the line for FRAP analysis from pole to pole. We draw it either to where DNA blocks movement of GFP (see *Figure 4—figure supplement 2a*) or far enough away from the bleach. It is important that the boundary conditions that are used in the analysis are still satisfied, meaning no transfer of particles over the ends of the line. We also make sure to only measure at places where there is no DNA present. We recorded the recovery in 150 time points with 750 ms exposure time for +25 GFP and 23–30 ms for −7 GFP. There was no additional time between exposures. The experiment was performed twice for both −7 and +25 GFP.

## Diffusion coefficients in the presence and absence of mRNA

We performed FRAP measurements over time of +25 GFP expressing *E. coli* cells with and without rifampicin treatment. The cells were grown in a tube with 4 mL LB (with 0.2 % v/v glycerol and 100 µg/mL ampicillin). We incubated the culture overnight at 30°C, with 200 rpm shaking. The next day we used 8 µL of the overnight culture to inoculate a 4 mL MBM culture (with 0.2 % v/v glycerol and 100 µg/mL ampicillin). Again we incubated the culture at 30°C, with 200 rpm shaking. The next morning, depending on the $OD_{600}$, the cultures were diluted with more MBM or allowed to continue growing. When the culture reached an $OD_{600}$ of 0.17–0.23 the production of +25 GFP was induced by adding 0.4 % w/v L-arabinose (from a 20 % w/v in MilliQ stock). At $OD_{600}$ of 0.30–0.34 we took 198 µL culture and added either 2 µL DMSO (control) or 2 µL DMSO with 50 mg/mL rifampicin (for a final concentration of 0.5 mg/mL). After mixing we deposited 4 µL sample on an APTES cover slide, put an object slide on top and the put the whole on the microscope stage. The stage was maintained at 30°C for the duration of the experiment. We measured the diffusion coefficients by FRAP and recorded the time for each measurement. We did each FRAP measurement on a unique cell. The replicates represent separate cultures on the day of the FRAP measurements.

## Sucrose gradient centrifugation of *E. coli* cell lysates and purified DNA

For each experiment on *E. coli* cell lysates, with *E. coli* expressing either −7 or +25 GFP, we deposited a small amount of cells from a glycerol stock into 4 mL LB medium (containing 0.2 % v/v glycerol and 100 µg/mL ampicillin). We incubated the cultures at 30°C with 200 rpm shaking overnight. The next day we took 20 µL of the LB culture to inoculate 10 mL MBM (0.28 Osm, osmolality adjusted with NaCl, containing 0.2 % v/v glycerol and 100 µg/mL ampicillin). Again we incubated overnight. The next morning the cultures had reached an $OD_{600}$ between 0.8–1.3, and were diluted to an $OD_{600}$ of 0.1–0.16 in 50 mL of fresh medium in a 250 mL flask. We added 0.4% (w/v) L-arabinose to induce protein expression. We incubated the cultures for 3.5–4 hr, to obtain an $OD_{600}$ of 0.25–0.37.

We centrifuged 44 mL of each culture at 5250 g, 20 min, 4°C. From this point onward we did all the work on ice and used cooled buffers. Pellets were suspended in 1 mL of resuspension buffer (20 mM Tris-HCl pH 7.5, 15 mM magnesium acetate, 100 mM ammonium acetate plus 6 mM 2-mercaptoethanol), resulting in around 200–300 fold dilution of cytoplasmic content. To calculate the dilution we assumed a cytoplasmic volume 0.5 fL (*Taheri-Araghi et al., 2015*) and $8 \times 10^8$ cells in 1 mL of culture of $OD_{600}$ of 1. To each tube containing the suspension we added around 0.2 mg of 106 µm glass beads (Sigma, St. Louis, MO, USA) and lysed the cells using two repetitions of 50 Hz oscillation for 5 min (TissueLyser LT, QIAGEN, Venlo, The Netherlands). We cooled the sample on ice in between repetitions. We added PMSF (100 mM in isopropanol stock) to the lysates to a final concentration of 1 mM. Then the lysates were centrifuged at 9000 g, for 2 min at 4°C. We took the supernatant and centrifuged it at 9000 g for 15 min at 4°C. We layered 800 µL of the resulting supernatant onto 8 mL of linear 10–40% sucrose gradient. The sucrose solutions contained 1 mM PMSF and were prepared with the resuspension buffer. We centrifuged the gradients using a swing-out rotor (SW 32.1 Ti, Beckman, Brea, CA, USA) at 125 000 g for 80 min at 4°C as described previously (*Maki et al., 2000*; *Puri et al., 2014*). We recorded a fluorescence profile over the sucrose gradient by dividing the gradient in 600 µL fractions, and measuring the fluorescence intensity for each fraction in a Jasco FP-8300 fluorimeter. We excited with 488 nm and recorded the emission from 500 to 600 nm (in 5 nm intervals). For the analysis we used the fluorescence emission at 510 nm. To correct for background fluorescence, we acquired spectra of 10, 20, 30% and 40% sucrose in resuspension buffer with 1 mM PMSF. A linear fit of the 510 nm emission intensities was used to calculate sucrose-caused background values for each fraction. For lysates of *E. coli* expressing −7 and +25 GFP the experiment was carried out two times.

For the samples containing DNA, we dissolved salmon testes DNA in autoclaved MQ to a final concentration of 1 mg/mL. The DNA was added to the cell lysates directly after the second centrifugation step; the sample was incubated for around 30 min on ice, before layering it onto the sucrose gradient. To determine the fractionation profile of DNA we layered 0.2 mg/ml DNA onto the linear sucrose gradient, except that the 2-mercaptoethanol and PMSF were omitted. The collected fractions were diluted 1:1 in 20 mM Tris-HCl pH 7.5, 15 mM magnesium acetate, 100 mM ammonium acetate. We determined DNA levels by measuring the absorbance of each fraction from 200 to 340 nm with 5 nm intervals, using Cary 100 Bio UV-VIS spectrometer. To correct for background absorption we measured fractions of pure MQ sample treated in the same way. For lysates of *E. coli* expressing +25 GFP with addition of DNA the experiment was carried out once for each DNA concentration; the DNA control experiment was also done once.

## Electron microscopy

We dialyzed the fractionated cell lysate samples in pre-cooled 20 mM Tris-HCl pH 7.5, 15 mM magnesium acetate, 100 mM ammonium acetate for 1 hr to remove the sucrose. The samples were pipetted on glow-discharged carbon-coated copper grids, excess liquid was removed by blotting and the grids were stained with 2% uranyl acetate. EM was performed on a Tecnai T20 electron microscope (FEI, Eindhoven, The Netherlands) operated at 200 kV, images were acquired with a 4000 SP 4K slow-scan CCD camera (Gatan, Pleasanton, CA, USA) as described previously (*Puri et al., 2014*).

## Purification of ribosomes

The protocol for isolating ribosomes is based on (*Blaha et al., 2000*) and (*Moazed et al., 1986*). For the isolation of ribosomes we used *E. coli* MC1061 cells. These cells harbor a pBAD vector (*Fulyani et al., 2013*). A small amount of cells from a glycerol stock was deposited into 30 mL of LB medium containing 100 µg/mL ampicillin in a 100 mL flask. We incubated the cultures at 37°C overnight with 200 RPM shaking. The next day we inoculated two cultures of 1 L fresh medium in 5 L flasks with 10 mL of the overnight culture each. We grew the cultures for 2–3 hr, until an $OD_{600}$ of 0.66–0.81. We centrifuged the cultures at 6000 g, 10 min, 4°C in a JLA-9.1000 rotor (Beckman). From this point onward we did all the work on ice and used cooled buffers. Pellets were suspended in 10–20 mL of resuspension buffer (20 mM HEPES-NaOH pH 7.5, 10 mM $MgCl_2$, 30 mM ammonium acetate, 150 mM KCl, 10 mM NaCl, 6 mM 2-mercaptoethanol, 1 mM PMSF). About 10 µg/mL of DNase I (Sigma) was added to the cell suspension. Cells were lysed by sonication for 12 min with

cycling on and off every 5 s. We used the Vibra-Cell VCX 130 sonicator (Sonics, Newtown, CT, USA) with 6 mm diameter probe at 70% amplitude, and we cooled the tube in an ethanol-ice bath. The lysates were centrifuged at around 30,000 g, 1 hr, 4°C in a MLA-80 rotor (Beckman). The top part of the supernatant (4 out of 6 mL in the tube) was carefully pipetted into clean centrifugation tubes. The supernatant was centrifuged at around 110,000 g, 17 hr, 4°C in a MLA-80 rotor. We discarded the supernatant, leaving an opaque, white-yellowish pellet. The pellet was washed gently with 1 mL of SEC buffer (20 mM HEPES-NaOH pH 7.5, 10 mM $MgCl_2$, 30 mM ammonium acetate, 150 mM KCl, 10 mM NaCl, 6 mM 2-mercaptoethanol). Then, we gently added 0.8–1.6 mL of SEC buffer onto the pellet and nutated for 1 hr at 4°C. The resulting suspension was centrifuged at 20,000 g, 7 min, 4°C to pellet the aggregates. We purified the resulting supernatant on a SEC setup (AKTA, GE Healthcare, Chicago, IL, USA, equipped with 1260 Infinity FLD fluorescence detector, Agilent, Santa Clara, CA, USA), using a Superdex 200 10/300 GL (GE Healthcare) column and SEC buffer as eluent. Three peak $A_{280}$ fractions were pooled, resulting in 1.5 mL of purified ribosomes. We measured absorbance at 260 and 280 nm, using a NanoDrop ND-1000 (Thermofisher, Waltham, MA, USA) spectrophotometer. Reported $A_{280}$ values varied between 0.27 and 1.08 at 1 mm light path, depending on the replicate, the amount of buffer used for resuspension of the ribosome pellet and the volume loaded into the SEC setup. $A_{260}/A_{280}$ ratio of replicates was constant at around 1.9. From now on, all mentioned absorbance values are reported with a 1 mm light path.

## Purification of −7 and +25 GFP

The protocol for isolating −7 and +25 GFPs is based on (*Thompson et al., 2012b*). For the purification of GFPs we used *E. coli* MG1655 strains containing pBAD bearing the gene for either −7 or +25 GFP. We deposited a small amount of cells in 30 mL of LB with 100 µg/mL ampicillin in a 100 mL flask for −7 GFP, and in 100 mL of medium in a 500 mL flask for +25 GFP. The cultures were incubated at 37°C overnight with 200 RPM shaking. The next day we inoculated 2 L and 6 L of fresh medium for −7 and +25 GFP, respectively, by adding 10 mL of overnight culture to 1 L of medium in a 5 L flask. At the time of inoculation, the protein expression was induced by adding L-arabinose to final concentration of 0.4 % w/v. We incubated the cultures at 30°C, 200 RPM for about 3 hr. We centrifuged the cultures at 6000 g, 10 min, 4°C in a JLA-9.1000 rotor. From this point onward we did all the work on ice and used cooled buffers. We suspended the pellets in GFP resuspension buffer (50 mM potassium phosphate pH 7.5, 2 M NaCl, 1 mM $MgCl_2$, 20 mM imidazole) at around 5–10 mL of GFP resuspension buffer per 1 L of culture. About 10 µg/mL of DNase I and about 10 µg/mL of RNase A (Sigma) were added to the cell suspension. Cells were lysed by sonication, for 12 min with cycling on and off every 5 s. We used Vibra-Cell VCX 130 sonicator (Sonics), with 6 mm diameter probe at 70% amplitude, and cooled the tube in an ethanol-ice bath. Immediately after the sonication, we added EDTA to a final concentration of 0.5 mM. We centrifuged the lysate at around 20,000 g, 20 min, 4°C in a JA-25.50 rotor (Beckman). We collected the supernatants and added Ni Sepharose 6 Fast Flow (GE Healthcare) resin, corresponding to 2 mL of column volume, to +25 and −7 GFP isolates each. The supernatants with the column material were nutated for 45 min at 4°C. Then we put the suspensions into columns, and washed each with 20 column volumes of GFP equilibration buffer (50 mM potassium phosphate pH 7.5, 2 M NaCl, 20 mM imidazole), then with 20 column volumes of GFP wash buffer (50 mM potassium phosphate pH 7.5, 2 M NaCl, 50 mM imidazole). The proteins were eluted with GFP elution buffer (50 mM potassium phosphate pH 7.5, 2 M NaCl, 500 mM imidazole), and around 1 mL of the highest intensity fractions were pooled. We then added EDTA to each pool to a final concentration of 5 mM. Purified −7 and +25 GFP were centrifuged at 20,000 g, 7 min, 4°C to pellet possible aggregates. We purified the resulting supernatant on a SEC setup (GE Healthcare AKTA equipped with Agilent 1260 Infinity FLD fluorescence detector), using Superdex 200 10/300 GL (GE Healthcare) column and SEC buffer (20 mM HEPES-NaOH pH 7.5, 10 mM $MgCl_2$, 30 mM ammonium acetate, 150 mM KCl, 10 mM NaCl, 6 mM 2-mercaptoethanol) as running buffer. We pooled three fractions with the highest 510 nm emission at 488 nm excitation into 1.5 mL of purified GFP solution for −7 and +25 GFP each. We measured the absorbance at 280 nm, using a NanoDrop ND-1000 (Thermofisher, Bleiswijk, NL) spectrophotometer. Reported $A_{280}$ values were around 0.015 for −7 GFP and 0.022 for +25 GFP, with little variation between replicates. Both −7 and +25 GFP solutions were visibly green.

LIFE Research article

emistry

## Analytical SEC experiments

During preliminary experiments we noticed aggregation occurring while mixing +25 GFP with ribosomes at low dilution of the SEC purification pools. The aggregation was reproducible throughout all replicates. Despite the aggregation not being visible by naked eye, we were still able to see a pellet after centrifugation when imaging the fluorescence of the tubes (*Figure 5e*). Through trial-and-error we established that mixing ribosomes at $A_{280}$ = 0.14 and +25 GFP at $A_{280}$ = 0.0013 results in no aggregation (visible by naked eye after centrifugation at 20,000 g, 7 min, 4°C), while producing a peak at 280 nm for ribosomes when run again on SEC.

For the main experiments ribosomes and GFPs were diluted in SEC buffer to $A_{280}$ = 0.14 and $A_{280}$ = 0.0013, respectively, in a total volume of 0.8 mL. The solutions were then centrifuged at 20000 g, 7 min, 4°C to pellet aggregates. In the cases where we mixed ribosomes with −7 or +25 GFP, the GFP was added to the ribosome solution. The solutions were incubated on ice for 10 min before centrifugation. We collected the supernatant and ran 0.6 mL immediately on SEC (GE Healthcare AKTA equipped with Agilent 1260 Infinity FLD fluorescence detector), using Superdex 200 10/300 GL (GE Healthcare) column and SEC buffer as running buffer, while monitoring absorption at 280 nm and emission at 510 nm with 488 excitation. The centrifugation tubes and remaining solution were imaged for fluorescence using the LAS-3000 (Fujifilm, Tokyo, Japan) imager. We performed all these measurements twice. Additionally, for +25 GFP we ran a ribosome/GFP mixture with a 1.8x higher GFP concentration ($A_{280}$ = 0.0024) that exhibited clear aggregation.

## Computational analysis of proteomes

We made histograms of the distributions of pI and net charge of all proteins encoded by the genomes of *E. coli*, *L. lactis* and *Hfx. volcanii*. All protein sequence data was retrieved from UniProt (*Bateman et al., 2015*). For *E. coli* we used a K12 strain (proteome ID: UP000000318), for *L. lactis* we used strain MG1363 (proteome ID: UP000000364) and for *Hfx. volcanii* we used strain DS2 (proteome ID: UP000008243). The *L. lactis* MG1363 strain is the parent strain of *L. lactis* NZ9000 that we used for FRAP (*Linares et al., 2010*). The *Hfx. volcanii* DS2 is the parent strain of *Hfx. volcanii* H1895 (*Strillinger et al., 2016*). We calculated the pI of each protein based on its amino acid sequence using the Isoelectric Point Calculator by Kozlowski (*Kozlowski, 2016*). We modified the program to allow for net charge calculations; the pI and net charge values we report are based on the IPC_protein $pK_a$ dataset of the Isoelectric Point Calculator. To calculate the net charge we used a pH of 7.5. To get the distributions in the cytoplasm we took only those proteins that have gene ontology labels cytoplasm and cytosol in the uniprot database(GO:0005737 and GO:0005829). For *E. coli* this yielded 1406 proteins (compared to 4254 proteins in the full genome), for *L. lactis* 253 (2383), and for *Hfx. volcanii* 177 (3987). For *E. coli* we also made pI and net charge distributions in which protein copy numbers are taken into account. To do this we took copy number data from Schmidt et al. (*Schmidt et al., 2016*). Specifically, we took the abundance data for *E. coli* BW25113 cultured in M9 glycerol.

We also made histograms of the distributions of pI and net charge of all proteins encoded by the genomes of *Buchnera aphidicola* (proteome ID: UP000001806), *Blochmannia floridanus* (proteome ID: UP000002192), *Onion yellows phytoplasma* (proteome ID: UP000002523), and *Wiggleworthia glossinidia brevipalpis* (proteome ID: UP000000562). For Buchnera the cytoplasmic fraction contained 190 proteins (compared to 572 proteins in the full genome), for Blochmannia 156 (583), for phytoplasma 87 (730), and Wigglesworthia 158 (617).

## Calculation of protein charge

To calculate the charge of the GFP variants we counted the number of Asp, Glu, Lys and Arg residues, used the $pK_a$ values of all (de)protonatable residues and used the Henderson-Hasselbalch equation to calculate the net charge. Results of these calculations give net charge 1–2 higher than net charge calculated using the modified IPC and the IPC_protein $pK_a$ dataset (−31.1, –8.2, −1.3, +5.6, +9.5, +9.5, +13.6, +23.5 for –30, –7, 0, +7, +11a, +11b, +15 and +25 GFP respectively). In reality, ions can specifically bind to proteins and thereby change the base net charge (*i.e.* before any ionic screening effects occur) (*Filoti et al., 2015*). This is especially true for anions (*Filoti et al., 2015*), which could affect our (quantitative) interpretation. Two examples: bovine serum albumin (measured charge, −13.8; calculated charge, −18.3) (*Filoti et al., 2015*) and hen egg white lysozyme

(measured charge, +5.1; calculated charge, +11) (*Gokarn et al., 2011*); the actual values depend on the type(s) of ion(s) present (*Gokarn et al., 2011*). We also assumed that each residue of a particular type (e.g. all aspartates) have the same $pK_a$ independent of context. To take ion binding and context dependent $pK_a$ values into account would be a whole study in itself.

## Derivation of *Equation 1*

For diffusion in one dimension the probability density for the position of a particle after time, $t$, is given by:

$$p(x) = \frac{N}{\sqrt{4\pi Dt}} e^{-\frac{x^2}{4Dt}} \qquad (7)$$

Here $x$ is the position, $p(x)$ is the probability as a function of $x$, $N$ is a normalization factor, $D$ is the diffusion coefficient, and $t$ is the time step. When the particle is free it moves with diffusion coefficient $D_{free}$ and when bound with $D_{bound}$. The particle goes back and forth between free and bound states a number of times in a certain period of time, $\Delta t$. Because the motion in each time step is independent of the other time steps, we can sum all time steps and distances travelled for the free state and we can do the same for the bound state. We end up with two equations like *Equation 7* but in one we have $D = D_{free}$ and $t = f_{free}\Delta t$ and in the other $D = D_{bound}$ and $t = (1 - f_{free})\Delta t$, with $f_{free}$ being the fraction of time that the particle is free. To get the probability density for the position of the particle after time step, $\Delta t$, we convolute the two equations. A convolution of a Gaussian function leads to another Gaussian in the following way:

$$f(x) * g(x) = \frac{N}{\sqrt{2\pi}\sigma_f} e^{-\frac{x^2}{2\sigma_f^2}} * \frac{N}{\sqrt{2\pi}\sigma_g} e^{-\frac{x^2}{2\sigma_g^2}} = \frac{N}{\sqrt{2\pi\left(\sigma_f^2 + \sigma_g^2\right)}} e^{-\frac{x^2}{2\left(\sigma_f^2 + \sigma_g^2\right)}} \qquad (8)$$

Here, $*$, is the symbol for a convolution. By comparing *Equations 7 and 8* we can see that $\sigma_f^2 = 2D_{free}f_{free}\Delta t$ and $\sigma_g^2 = 2D_{bound}(1 - f_{free})\Delta t$. We can also define an effective diffusion coefficient, $D_{eff}$, such that $\sigma_f^2 + \sigma_g^2 = 2D_{eff}\Delta t$. Combining the last results and dividing by $2\Delta t$ we obtain *Equation 1*.

This derivation depends on six assumptions, as described in the discussion. The justification for these assumptions are as follows. We assumed that (1) the exchange between free and bound state is much faster than the FRAP measurement. This is justified because we did not observe two-component recovery in the FRAP data. We assumed that (2) the highest diffusion coefficient of all variants in a given organism reflects the free state of GFP; justified by the fact that for *E. coli* the GFP diffusion coefficient levels off towards more negative charge. We assumed that (3) GFPs bind solely to ribosomes; this we have shown experimentally. We assumed that (4) the total number of binding sites on all ribosomes is higher than the number of GFPs. The first justification is that the number of binding sites is about $10^6$ and it is unlikely that we express that many GFPs because the cell only has about $3 \times 10^6$ total proteins. The second justification is that 99% of +25 GFP appears to be bound, meaning that there are enough binding sites to bind all GFPs. We assumed that (5) the decrease of diffusion coefficient with net positive charge has the same origin in all three organisms. This assumption is plausible because all three organisms have ribosomes, and assuming the same cause for slow diffusion is the most parsimonious. Finally, we assumed that (6) the ribosome diffusion coefficient is the same in all three organisms. Here the justification is that for all three organisms $-7$ GFP diffusion coefficients are similar, suggesting similar crowding, for *E. coli* and *L. lactis* the diffusion coefficient are similar also for a big protein complex (*Mika et al., 2014*). The consequence for violating assumption one is that there is no $D_{eff}$ to speak of and the whole calculation becomes irrelevant. The consequence for violating assumptions 2, 3, 5, and 6 is that the numerical values coming out of the equation will be different, with the severity of the error depending on the difference in diffusion coefficients. The consequence of violating assumption four is that there will be free GFP irrespective of affinity as there are no more binding sites to fill.

## Acknowledgements

The work was funded by a NWO TOP-PUNT and ERC Advanced grant (ABCVolume) to BP. We acknowledge Thorsten Allers for providing us with the strain, plasmid and protocols for *Hfx. volcanii*; David Liu and David Thompson for providing us with surface modified GFPs; Christiaan M Punter for programming scripts for bioinformatic analysis; Gea K Schuurman-Wolters for help with GFP and ribosome purification, and SEC experiments; Marc CA Stuart for EM imaging; and Fangfang Guo for help with some of the FRAP measurements.

## Additional information

### Funding

| Funder | Grant reference number | Author |
| --- | --- | --- |
| H2020 European Research Council | Advanced Grant (ABC Volume) | Bert Poolman |
| Nederlandse Organisatie voor Wetenschappelijk Onderzoek | TOP-PUNT.13.006 | Bert Poolman |

The funders had no role in study design, data collection and interpretation, or the decision to submit the work for publication.

### Author contributions

Paul E Schavemaker, Conceptualization, Data curation, Formal analysis, Validation, Investigation, Methodology, Writing—original draft; Wojciech M Śmigiel, Conceptualization, Data curation, Formal analysis, Investigation, Methodology, Writing—review and editing; Bert Poolman, Conceptualization, Resources, Formal analysis, Supervision, Funding acquisition, Validation, Writing—review and editing

### Author ORCIDs

Wojciech M Śmigiel http://orcid.org/0000-0002-1868-8689
Bert Poolman http://orcid.org/0000-0002-1455-531X

### Decision letter and Author response

Decision letter https://doi.org/10.7554/eLife.30084.035
Author response https://doi.org/10.7554/eLife.30084.036

## Additional files

### Supplementary files

• Supplementary file 1. GFP sequences and diffusion coefficients (A) Amino acid sequences of the GFP variants. (B) Diffusion coefficients in *E. coli*, *L. lactis* and *Hfx. volcanii*. (C) *P*-values for pairwise comparisons of diffusion coefficients for GFP variants in *E. coli* (Eco), *L. lactis* (Lla), and *Hfx. volcanii* (Hvo). For *E. coli* we also compared diffusion coefficients under normal (0.28 Osm) and shock conditions (1.2 Osm). (D) Fitting parameters for the relation between diffusion coefficient, GFP net charge, and ionic strength.
DOI: https://doi.org/10.7554/eLife.30084.018

• Transparent reporting form
DOI: https://doi.org/10.7554/eLife.30084.019

### Major datasets

The following previously published datasets were used:

| Author(s) | Year | Dataset title | Dataset URL | Database, license, and accessibility information |
|---|---|---|---|---|
| Hayashi K, Morooka N, Yamamoto Y, Fujita K, Isono K, Choi S, Ohtsubo E, Baba T, Wanner BL, Mori H, Horiuchi T | 2006 | E. coli K12 proteome | http://www.uniprot.org/proteomes/UP000000318 | Publicly available at UniProt (proteome ID: UP000000318) |
| Wegmann U, O'Connell-Motherway M, Zomer A, Buist G, Shearman C, Canchaya C, Ventura M, Goesmann A, Gasson MJ, Kuipers OP, van Sinderen D, Kok J | 2007 | L. lactis sub. cremoris MG1363 proteome | http://www.uniprot.org/proteomes/UP000000364 | Publicly available at UniProt (proteome ID: UP000000364) |
| Hartman AL, Norais C, Badger JH, Delmas S, Haldenby S, Madupu R, Robinson J, Khouri H, Ren Q, Lowe TM, Maupin-Furlow J, Pohlschroder M, Daniels C, Pfeiffer F, Allers T, Eisen JA | 2010 | Hfx. volcanii DS2 proteome | http://www.uniprot.org/proteomes/UP000008243 | Publicly available at UniProt (proteome ID: UP000008243) |
| Shigenobu S, Watanabe H, Hattori M, Sakaki Y, Ishikawa H | 2000 | Buchnera aphidicola subsp. Acyrthosiphon pisum (strain APS) proteome | http://www.uniprot.org/proteomes/UP000001806 | Publicly available at UniProt (proteome ID: UP000001806) |
| Gil R, Silva FJ, Zientz E, Delmotte F, González-Candelas F, Latorre A, Rausell C, Kamerbeek J, Gadau J, Hölldobler B, van Ham RC, Gross R, Moya A | 2003 | Blochmannia floridanus proteome | http://www.uniprot.org/proteomes/UP000002192 | Publicly available at UniProt (proteome ID: UP000002192) |
| Oshima K, Kakizawa S, Nishigawa H, Jung HY, Wei W, Suzuki S, Arashida R, Nakata D, Miyata S, Ugaki M, Namba S | 2004 | Onion yellows phytoplasma (strain OY-M) proteome | http://www.uniprot.org/proteomes/UP000002523 | Publicly available at UniProt (proteome ID: UP000002523) |
| Akman L, Yamashita A, Watanabe H, Oshima K, Shiba T, Hattori M, Aksoy S | 2002 | Wigglesworthia glossinidia brevipalpis proteome | http://www.uniprot.org/proteomes/UP000000562 | Publicly available at UniProt (proteome ID: UP000000562) |

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
