## [Decision Letter]

Thank you for submitting your article "Ribosome surface properties impose limits on the nature of the cytoplasmic proteome" for consideration by *eLife*. Your article has been favorably evaluated by Ian Baldwin (Senior Editor) and three reviewers, one of whom is a member of our Board of Reviewing Editors. The following individual involved in review of your submission has agreed to reveal their identity: Conrad W Mullineaux (Reviewer #2).

The reviewers have discussed the reviews with one another and the Reviewing Editor has drafted this decision to help you prepare a revised submission.

Summary:

In their manuscript "Ribosome surface properties impose limits on the nature of the cytoplasmic proteome" the authors have measured diffusion of a set of GFP variants with different net charge in the bacterial cytoplasm. They show that the positively charged proteins diffuse much slower than the negative ones and propose that this behavior arises from protein binding to the negatively charged ribosomes. The authors further hypothesize that such binding to ribosomes imposes evolutionary selection against positively charged cytoplasmic proteins in bacteria, possibly explaining why such proteins are rare. Finally, they discuss the case of several endosymbiotic bacteria, which apparently contain a much larger fraction of positively charged proteins. The findings of the manuscript have potentially fundamental biological importance.

Essential revisions:

The reviewers raised several concerns related to the analysis and interpretation of the data and suggested additional experiments in order to further support authors' conclusions. These concerns must be addressed before the manuscript can be accepted.

1) Although binding to ribosomes is a plausible explanation for the observed slow diffusion of the positively charged GFP variants, it should be further confirmed by direct measurements of protein binding to purified ribosomes, using for example ITC. You should perform these experiments with both positively and negatively charged GFP constructs under well-defined physiological conditions of pH and ionic strength.

2) Further evidence is required to rule out that the slower diffusion of positively charged proteins is not due to their binding to mRNA. This could also be done using *in vitro* experiments as described above.

3) The discussion of protein charge distribution in endosymbionts is too long and speculative, as the authors have no experimental data for these species to support their hypothesis. This discussion should be substantially shortened.

4) The claim made in the title of the manuscript may be too strong, since the authors could not directly prove that binding to ribosomes was indeed the main reason for evolutionary selection against positively charged proteins. Toning down the title to something like "Ribosome surface properties likely impose limits on the nature of the cytoplasmic proteome" as well as rephrasing of the Abstract seem appropriate.

*Reviewer #1:*

The manuscript by Schavemaker et al. describes vary interesting observation that positively charged GFP variants show much slower diffusion in bacterial cytoplasm. This finding is novel, and it potentially has fundamental biological importance. Authors further propose that this effect could be explained by the binding of positively charged proteins to ribosomes. However, there are several serious issues related to the analysis and interpretation of the data that need to be addressed before the manuscript could be considered for publication.

1) The central claim of the manuscript is that slower diffusion of the positively charged GFP is primarily due to its binding to the surface of ribosomes. In this context, I find the interpretation of Figure 4 questionable. Authors show that treatment with rifampicin results in faster diffusion of +25 GFP, although even upon such treatment the diffusion coefficient remains well below one of -30 GFP. As far as I understand their logic, the authors assume that this increase is due to dissociation of ribosomes bound to +25 GFP from mRNA (which becomes degraded upon rifampicin treatment). Couldn't the same effect be explained by binding of +25 GFP to mRNA itself? This would not rule out that binding to ribosomes is an important factor in limiting diffusion positively charged proteins in bacteria. But unless the authors can exclude direct binding to mRNA, it may have as much effect as the binding to ribosomes, meaning that the title and the Abstract should be formulated more cautiously.

2) Subsection “GFP variants and histograms of diffusion coefficients of surface modified variants of GFP in *Escherichia coli*”, end of last paragraph. The authors state that they did not find association of +25 GFP with the membrane, without presenting any evidence. What I find worrying is that, in Figure 2—figure supplement 1, in some "very heterogeneous" cells +25 GFP appears to localize to the cell periphery, which might indicate binding to the membrane. Although such cells were excluded from further analysis, I think it is possible that even in cells without obvious peripheral localization some membrane binding occurs – unless the authors have some direct evidence that +25 GFP does not bind to the membrane.

3) In using the equation 1 that describes diffusion with binding, the authors make a number of assumptions. Although the assumptions are clearly specified (and plausible), I think it would be important to justify these assumptions more explicitly (e.g., assumption #1) and mention how would violation of these assumptions affect the conclusions.

4) Figure 2 and Figure 3: why is GFP with zero charge not shown?

*Reviewer #2:*

I enjoyed reading this paper, which introduces a really fresh and exciting approach to thinking about the factors that influence the effective viscosity of the cytoplasm. The idea that the majority of bacterial cytoplasmic proteins could be negatively charged to minimise interaction with ribosomes and maximise their rates of diffusion was a revelation to me. The paper reports an incisive series of experiments that provide convincing evidence that positive surface charge slows diffusion in the cytoplasm and this effect is at least largely due to interaction with ribosomes bound to RNA. The paper is well-written, and the data are well-presented. However, the authors do need to calculate p- values to confirm significant differences between the diffusion coefficients for different GFP variants and under different conditions. Otherwise the only part of the manuscript that I found unsatisfactory was the discussion of the positively-charged proteome of 4 endosymbiotic species (Abstract; Figure 7—figure supplement 2, subsection “Proteome analysis reveals potentially slow proteins”, last two paragraphs). This section raises far more questions than it answers. Here there is no accompanying data in whether cytoplasmic protein diffusion actually is slower in these species, whether the cytoplasmic pH or ionic strength is different, or whether or not the positive proteome is a general feature of endosymbionts (what about mitochondria and chloroplasts too?). And are there non-endosymbiotic bacteria out there with a positively-charged proteome? I would recommend reducing this section to a brief comment that are certain exceptions to the rule of a predominantly negatively-charged cytoplasmic proteome, that deserve further investigation in future.

*Reviewer #3:*

In this study, the authors have estimated the lateral diffusion coefficients (D) of a set of surface modified GFPs and have shown that in *E. coli* D depends on the net charge and its distribution over the protein. They show that the positively charged proteins diffuse much slower than the negative ones and this behavior probably arise from their nonspecific association to the negatively charged ribosomes in the cytoplasm. Thus they claim that the presence of limited number of positively charged proteins in *E. coli* is most probably for avoiding their crowding on the ribosome – the protein synthesis machinery of the cell. The study is certainly interesting and makes a big claim, but some results need to be clarified and established by additional experiments in order to make such conclusion.

1) The nucleoid occlusion of +25 GFP is not obvious in majority of the cells shown in the Figure 4—figure supplement 1. Why is it so?

2) Also, only 9 out of 46 cells (only 20%) showed shrinking of the nucleoid upon Chl treatment. This is somewhat strange and can be certainly improved. 1.5 h incubation may be too long; different time period of incubation should be tried to obtain higher fraction of the cells with typical chloramphenicol features.

3) The ribosomal association of the +25 GFP is shown by co-sedimentation with the ribosomes in sucrose gradient analysis. Further the Kd has been determined by indirect method as 6.7 uM. Since this data is the key to the whole study, I strongly recommend that they validate it with other method such as ITC. In fact, the authors should subject all their GFP constructs to ITC analysis for estimating affinity to the (purified) ribosomes and plot the Kd values against the net charge of the GFPs. This will allow better establishment of a link with the proteome analysis already presented here.

4) The title of the article must be changed to match the exact finding of the study. It makes an exaggerated claim that the surface property of the ribosome limits the nature of the cytoplasmic proteome, for which the authors have no direct evidence whatsoever! It will require evolution experiments where changing surface charge of the ribosome will allow spontaneous inclusion of the positive charge in various proteins. It is not even discussed adequately and mentioned in the Abstract with uncertainty 'Ribosome surface properties MAY thus limit…..'.

5) The referencing is not done properly. For ribosomal nucleoid exclusion, and also, for changes in the cell shape, ribosome and nucleoid distribution due to chloramphenicol and cephalexin treatment the authors should refer to Chai et al., JBC, 2014.

---

## [Author Response]

Essential revisions:The reviewers raised several concerns related to the analysis and interpretation of the data and suggested additional experiments in order to further support authors' conclusions. These concerns must be addressed before the manuscript can be accepted.1) Although binding to ribosomes is a plausible explanation for the observed slow diffusion of the positively charged GFP variants, it should be further confirmed by direct measurements of protein binding to purified ribosomes, using for example ITC. You should perform these experiments with both positively and negatively charged GFP constructs under well-defined physiological conditions of pH and ionic strength.

We have purified ribosomes and have analysed their ability to bind purified -7 and +25 GFP, using size exclusion chromatography. The pH and ionic strength reflect those in *E. coli*. We find that +25 GFP co-migrates with ribosomes whereas -7 GFP does not, confirming what we had found in the *in vivo* experiments and on cell extracts. The new data are shown in Figure 5, four additional panels. ITC is not possible because ribosomes are predicted to have multiple binding sites for +25 GFP and there will not be a unique dissociation constant, hence the ITC data will not be interpretable.

2) Further evidence is required to rule out that the slower diffusion of positively charged proteins is not due to their binding to mRNA. This could also be done using in vitro experiments as described above.

Ribosome diffusion coefficients have been determined (by others) in the presence and absence of mRNA showing an increase in diffusion coefficient, from 0.04 to 0.6 µm^2^/s upon removal of mRNA (Bakshi et al. 2012). When we remove the mRNA, by the same method, in our diffusion measurements with +25 GFP we see a (similar) increase in diffusion coefficient from 0.14 to 0.5 µm^2^/s. There is not a lot of room for mRNA to do anything here as most of the change in D is already explained by change in the ribosome diffusion coefficient.

We do not rule out that +25 GFP binds to mRNA to some extent. Similarly, we find no evidence for association of +25 GFP to DNA or the membrane, and thus conclude that these are not major factors. We have made this clearer in our revised manuscript. If +25 GFP does bind to mRNA to some extent this wouldn’t take away from our claim that ribosome surface properties affect the diffusion coefficient of +25 GFP, and thus may put limits on the nature of the proteome. This simply means that other factors may also put limits on the nature of the proteome.

3) The discussion of protein charge distribution in endosymbionts is too long and speculative, as the authors have no experimental data for these species to support their hypothesis. This discussion should be substantially shortened.

We have shortened the endosymbiont discussion by more than twofold. However, there is no speculation in our discussion on endosymbionts. We merely state some facts that appear paradoxical. Surely this paradox will be resolved in time. We offer this simply as a problem to be solved.

4) The claim made in the title of the manuscript may be too strong, since the authors could not directly prove that binding to ribosomes was indeed the main reason for evolutionary selection against positively charged proteins. Toning down the title to something like "Ribosome surface properties likely impose limits on the nature of the cytoplasmic proteome" as well as rephrasing of the Abstract seem appropriate.

We agree with this point. In fact we already had the weaker claim present in the abstract. Now we have adjusted the title, and the conclusion, so that they also contain the weaker claim.